



# Forestation tends to create favourable conditions for convective precipitation in the Mediterranean Basin

Jolanda J.E. Theeuwen[1,2], Sarah N. Warnau[2,3], Imme B. Benedict[3], Stefan C. Dekker[1], Hubertus (Bert) V.M. Hamelers[2,4], Chiel C. van Heerwaarden[3], Arie Staal[1].

[1]Copernicus institute of sustainable development, Utrecht University, Utrecht, 3584CB, the Netherlands
[2]Wetsus, European centre of excellence for sustainable water technology, Leeuwarden, 8911MA, the Netherlands
[3]Meteorology and Air quality group, Wageningen University & Research, Wageningen, 6708PB, the Netherlands
[4]Environmental technology group, Wageningen University & Research, Wageningen, 6708PB, the Netherlands

Correspondence to: Jolanda J.E. Theeuwen (j.j.e.theeuwen@uu.nl)

**Abstract.** The Mediterranean Basin is identified as a climate change hotspot and prone to future drying. Previous studies indicate that the effect of forests on precipitation remains unclear for the Mediterranean Basin specifically. Here we use a simple model to simulate the development of the atmospheric boundary layer (ABL) to determine the impact of forest on convective rainfall potential. There is convective rainfall potential when (1) the ABL reaches the lifting condensation level, and (2) there is sufficient convective available potential energy. We model the ABL development over the Mediterranean Basin covered fully with bare soil and forest to determine its land cover sensitivity. In addition, we examine the sensitivity of the ABL to variations in soil moisture for the forest scenario specifically. We identify two distinct responses to forestation in the Mediterranean Basin dependent on soil moisture content. Forestation contributes to warming and drying in relatively dry regions (low soil moisture content) and to cooling and wetting in relatively wet regions (high soil moisture content), indicating that dry gets drier and wet gets wetter. We find that both forestation and an increase in soil moisture can contribute to convective rainfall potential. In regions with a relatively high soil moisture content, forestation positively influences both the convective available potential energy, and the crossing of the ABL and lifting condensation level. The results show that forestation in the Mediterranean Basin may reduce future drying in relatively wet regions and enhance future drying in relatively dry regions.

## 1 Introduction

Unsustainable land use and global warming lead to water scarcity and desertification in different regions across the globe (Bestelmeyer et al., 2015; Intergovernmental Panel on Climate Change (IPCC), 2023). Especially Mediterranean-type climate regions are identified as climate change hotspots (Ali et al., 2022; Diffenbaugh et al., 2008; Döll, 2009; Fraser et al., 2013) and are prone to drying (Pokhrel et al., 2021). In those regions, forestation may increase freshwater availability when the increase in evapotranspiration promotes precipitation (Cui et al., 2022). It is expected that forestation specifically may enhance rainfall in dry regions due to the buffering effect of Mediterranean forests on precipitation (O'Connor et al., 2021) as deep roots make deep soil moisture available during dry periods (Brunner et al., 2015). However, increased





evapotranspiration can also reduce streamflow (Galleguillos et al., 2021) and therefore, forestation should be done strategically, such that it contributes to rainfall enhancement (Staal et al., 2024b), and not to local drying.

Rainfall enhancement through forestation has more potential in regions characterized by high atmospheric moisture recycling ratios (Hoek van Dijke et al., 2022; Tuinenburg et al., 2022). Atmospheric moisture recycling describes the return of evaporated water over land and can be studied at different spatial scales. Local evaporation recycling, which is the return of evaporated water as precipitation locally, reduces local drying as evaporative losses are partially compensated for by rainfall (Theeuwen et al., 2023). The Mediterranean Basin shows most potential to enhance local rainfall with regreening

compared to all other Mediterranean-type climate regions, particularly during summer when the local recycling ratio is largest (Theeuwen et al., 2024). As the amount of evaporated water that recycles is also influenced by regreening, regreening could potentially increase moisture recycling and rainfall in regions that currently have low local recycling ratios.

Atmospheric moisture recycling is often calculated using historical weather data (Van der Ent et al., 2014; Hoek van Dijke et al., 2022; Tuinenburg et al., 2022) even though a change in land cover is expected to affect the recycling ratio through

changes in land-atmosphere interactions. A limited number of studies combines an Earth system model with a moisture tracking model to calculate the moisture recycling ratio for different land cover scenarios (De Hertog et al., 2024; Staal et al., 2024a). In such a set-up, remote impacts on local precipitation cannot be excluded (De Hertog et al., 2024). To isolate the local effects of changes in land cover on local precipitation a different model approach is necessary. In this research we explore if forestation creates favourable atmospheric conditions for local convective precipitation through changes in local

land-atmosphere interactions for the Mediterranean Basin.

First, we focus on how changes in the energy balance and evaporation due to land use changes affect the development of the atmospheric boundary layer (ABL) and the level at which water vapor starts to condense (i.e., lifting condensation level, LCL). The ABL is the lower part of the atmosphere that grows from ~100 m to several kilometres during the day due to the release of thermal heat at the Earth's surface. When the ABL crosses the LCL, water vapor in the ABL starts to condense

and convective clouds can develop. Second, we focus on the convective available potential energy (CAPE). Sufficient CAPE ($\geq 400$ Jkg$^{-1}$) needs to be available for the development of deep convective clouds that can produce precipitation (Yin et al., 2015). When both the ABL and LCL cross and the CAPE is at least 400 Jkg$^{-1}$ there is convective rainfall potential.

It is expected that forestation enhances convective rainfall potential due to an increase in ABL height, a decrease in LCL height and an increase of CAPE. Due to the lower albedo of forests compared to bare soil the net surface radiation increases

(Fig. 1), which enhances the latent and sensible heat fluxes (Bonan, 2008). An increase in these surface fluxes is beneficial for the deepening of the ABL (Van Heerwaarden and Teuling, 2014) as well as the development of CAPE (Yin et al., 2015). In addition, evapotranspiration is expected to increase due to forestation (Fig. 1). The larger amount of moisture present in the atmosphere reduces the LCL height (Yin et al., 2015). These changes are beneficial to the crossing of the ABL and LCL as well as to reaching a sufficient amount of CAPE.

To simulate the development of the ABL throughout the day we use a model in which the ABL is represented as a homogenous "slab" with uniform properties. We determine the impact of forestation in the Mediterranean Basin on the



development of the ABL, the LCL and CAPE to investigate under which conditions the ABL and LCL cross and sufficient CAPE is available to trigger rainfall. We simulate the ABL development in the entire Mediterranean Basin for the months May and June. To study the impact of forestation across the Mediterranean Basin, our study compares the occurrence of

convective rainfall potential over bare soil and forest.

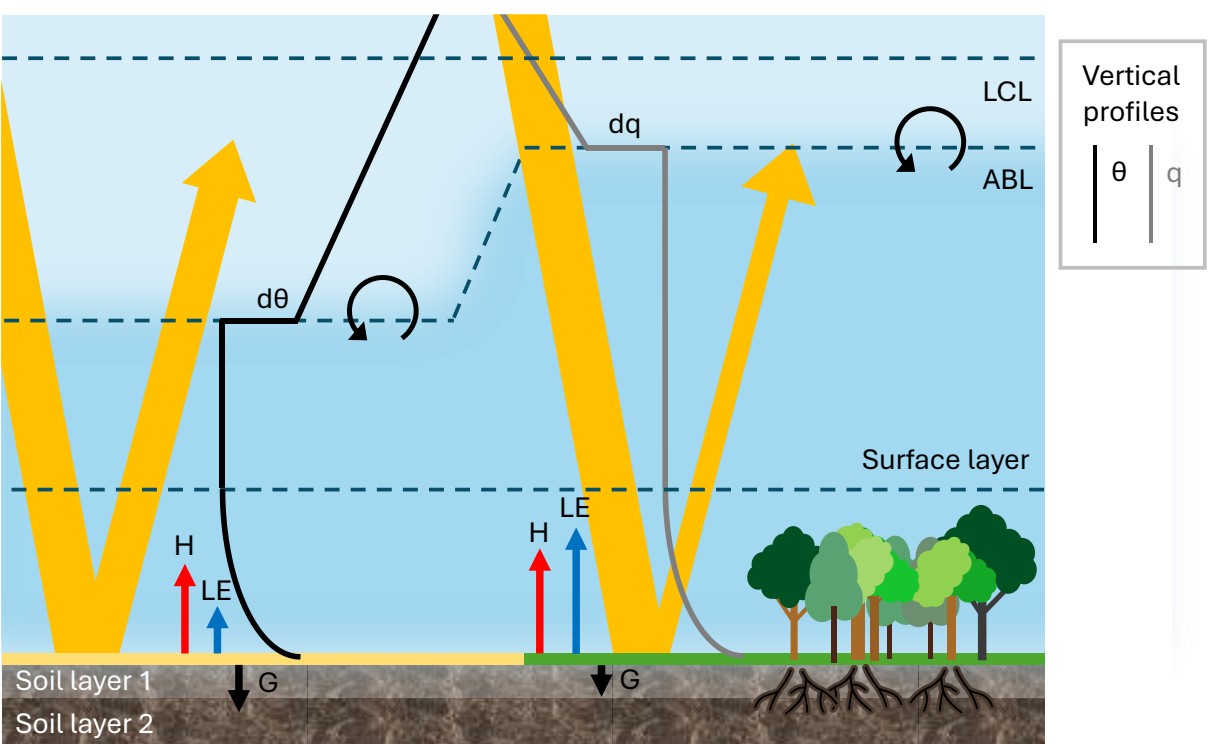

**Figure 1: Conceptual model that describes the vertical potential temperature (black) and specific humidity (grey) profiles of the atmospheric boundary layer (ABL) and the ABL development over two different land cover types (left: bare soil, right: forest). For the ABL over bare soil only the vertical temperature profile is shown and for the ABL over forest only the vertical humidity**
**profile is shown. θ: potential temperature, q specific humidity, LCL: lifting condensation level, BLH: boundary layer height, dθ: jump in θ at the top of the ABL, dq: jump in q at the top of the ABL, H: sensible heat flux, LE: latent heat flux, G: ground heat flux. The circular arrows at the top of the ABL indicate entrainment from dry and warm air into the ABL. The yellow arrows indicate the incoming solar radiation and the part that is reflected back towards the atmosphere, which varies due to the different albedo of the two land cover types. Throughout the day the ABL deepens. Clouds can develop when the ABL and LCL cross.**

**2 Methods**

We use the Chemistry Land-surface Atmosphere Soil Slab (CLASS) model (van Heerwaarden et al., 2010; Vilà-Guerau de Arellano et al., 2015) to study the impact of land cover changes and soil moisture availability on the potential for convective rainfall in the Mediterranean Basin in May and June for the years 2013-2022. This simulation is done for early summer as



this is the start of the dry season. Using this model we simulate the development of the atmospheric boundary layer (ABL)
and lifting condensation level (LCL) throughout the day (between 6 AM and 3 PM) for two land cover scenarios and five
soil moisture scenarios. In this section, we describe the study region, CLASS model, input data, sampling of input data,
experimental design, postprocessing, validation and analysis of the model output.

## 2.1 The Mediterranean Basin

The study region, the Mediterranean Basin, includes all regions located around the Mediterranean Sea that have a
Mediterranean climate according to the Köppen climate classification (Fig. A1). Some additional small areas with a semi-
arid climate were included to minimize fragmentation of the study region. The Mediterranean Basin is the largest of five
major Mediterranean-type climate regions globally. A Mediterranean climate is characterized by wet winters with mild
temperatures and dry and hot summers (Esler et al., 2018). Although precipitation falls predominantly in winter, during late
spring and summer there is convective precipitation in the region (Treppiedi et al., 2023; Wallace and Hobbs, 2006). The
region has multiple mountains with peaks up to 2400 m that promote local moisture recycling (Theeuwen et al., 2024).
Throughout the Mediterranean Basin the soil moisture content varies between 0 and 0.4 m3/m3  (Fig. A1). The study region
consists of 2868 grid cells of 0.25°x0.25°, which means that it spans a distance of approximately 4850 km from north to
south and approximately 1400 km from west to east. As the region is located in several time zones, we refer to the local solar
time of each specific grid cell throughout the manuscript.

## 2.2 The CLASS model

The CLASS (Chemistry Land-surface Atmosphere Soil Slab) model is based on multiple models that describe the land and
atmosphere (van Heerwaarden et al., 2010; Vilà-Guerau de Arellano et al., 2015). The soil moisture and heat transport in and
out of the soil are described with a two layer force-restore soil model. Force refers to the external inputs that affect soil
moisture and temperature; restore refers to the intrinsic properties of the soil (Noilhan and Mahfouf, 1996). A slab model is
used to simulate the development of the convective ABL (Tennekes, 1973; Tennekes and Driedonks, 1981). During the day,
the ABL is well mixed and potential temperature and specific humidity within this layer can be represented by a single
value, i.e., slab (Vilà-Guerau de Arellano et al., 2015). The exchange of heat and moisture between the soil and the
atmosphere is regulated by a surface energy balance where the Penman-Monteith equation is used for closure (Monteith,
1965). The modelled atmospheric surface layer is based on the Monin–Obukhov similarity theory (Monin and Obukhov,
1954). In our set up, we used the land surface, radiation, mixed layer, shear growth, and surface layer modules. In addition,
we used the land surface parameterization Jarvis-Stewart (Jarvis, 1976). Advection fluxes of heat and moisture were
prescribed and large-scale subsidence was neglected. Finally, the integration timestep was 15 seconds and output was
collected in 15 minute interval means. The code for CLASS is available through http://classmodel.github.io/. Previous
studies with CLASS that used observations for validation showed that the model reproduces the ABL processes well (Van
Heerwaarden and Teuling, 2014; Wouters et al., 2019).



### 2.3 Model input

For our study, the input data for CLASS are divided into parameters and variables from which the latter group is split into variables that can directly be retrieved from data and variables that we needed to calculate (Table A1). The equations that we used to calculate these variables are presented in Appendix B. Variables were directly obtained from ERA5 data (Hersbach et al., 2020) or calculated from ERA5 data. We use ERA5 data as it provides the best available spatial and temporal coverage of the study region. However, the accuracy of the spatial and temporal variations in ERA5 surpasses the accuracy of its absolute values (Hersbach et al., 2020). Therefore, we need to carefully interpret our results and should mainly focus on spatial patterns as the absolute changes in the different variables are likely less meaningful. We used hourly ERA5 data, both "at pressure levels" and "at a single level" (Table A1). All parameter values were obtained from the ECMWF IFS documentation (ECMWF, 2010).

A typical ABL height in the morning is in the order of tens of meters to several hundred meters (Stull, 1988), so, first, we assumed that at 6 AM local time the boundary layer has a height of 100 m and is shear-mixed. Second, to calculate the soil moisture content for the soil sensitivity scenarios we used wilting point, field capacity and saturation for a medium textured soil as the most common soil types in the Mediterranean region are Cambisols (26%), Calcisols (22%), Leptosols (20%), and Luvisols (10%) (Allam et al., 2020), which vary from coarse to fine-textured. We assume that wilting point and field capacity are constant throughout the study region. Third, for the forest scenario we assumed a typical LAI for a mixed forest (LAI = 5 m2/m2). Finally, for the albedo we assumed that the bare soil is light colored, which maximizes the difference between the albedo of a forest and albedo of bare soil.

To calculate the atmospheric conditions, we assumed a constant air density within the ABL and hydrostatic equilibrium. Furthermore, we assumed idealized linear profiles for the specific humidity and potential temperature.

### 2.4 Sampling

Due to the large number of days and grid cells in the study period and region we designed a specific sampling method for our input data that ensures a distribution that represents the entire Mediterranean Basin during this period. For each grid cell we ran the model 20 times. Each year was sampled twice, where the exact days were obtained randomly. This resulted in a total amount of 57,360 samples. For each of these samples we ran the model using the specific input data for that location and day.

### 2.5 Experimental design

To study the influence of land cover and soil moisture on the LCL and the development of the ABL we designed two land cover scenarios and five soil moisture scenarios (Table 1). The impact of forestation is studied within grid cells specifically and does not affect the interactions between grid cells. For the land cover scenarios we vary the vegetation fraction, LAI, albedo, aerodynamic resistance, and roughness length for heat and momentum (Table A1). For the land cover scenarios, soil



moisture varies among the grid cells and is obtained from ERA5. For the soil moisture scenarios the land is covered in forest. In these scenarios soil moisture is constant throughout the Mediterranean Basin for each simulation, however, soil moisture varies among different simulations, ranging from wilting point (WP) to field capacity (FC) (Table 1).

**Table 2: Experimental set-up describing the land cover and soil moisture scenarios. WP: wilting point and FC: field capacity. A more detailed description of the differences in input data for the bare soil runs and forest runs can be found in Table A1.**

| | Soil moisture | Land cover |
|---|---|---|
| **Land cover sensitivity scenarios** | | |
| Bare soil | ERA5 data | Bare soil |
| Forest | ERA5 data | Forest |
| **Soil moisture sensitivity scenarios** | | |
| Low | WP | Forest |
| Medium-low | WP+1/4(FC-WP) | Forest |
| Medium | WP+1/2(FC-WP) | Forest |
| Medium-high | WP+3/4(FC-WP) | Forest |
| High | FC | Forest |

## 2.6 Postprocessing

To account for extreme values in our output data due to inconsistencies that are part of the input data, we filter out unrealistic

output of CLASS by filtering the values of soil moisture of the top layer, relative humidity of the ABL, ABL height, potential temperature, and the jumps of potential temperature and specific humidity at the top of the ABL. The filters truncate the distribution for each of the variables and remove samples with values from below the 5th percentile and above the 95th percentile for all variables except the ABL height. As the ABL was not normally distributed we removed the simulations in which the ABL did not grow (≤100 m) and the variables above the 85th percentile. The exact filters can be

found in Table A2. 29957 Samples (52%) pass the filter for both land cover type scenarios and these samples are used to study the land cover sensitivity. Due to the poor quality of the ERA5 data, specifically over dry regions, it was expected that there would be an error in the model output. The larger uncertainty in the ERA5 data for drier regions (https://confluence.ecmwf.int/display/CKB/ERA5, accessed 16-12-2024) explains why, after sampling, there are fewer samples available in relatively dry regions, i.e., low soil moisture, than in wet regions (Fig. A2 and A3). After this

postprocessing step, each grid cell had on average $10 \pm 5$ samples and there are 69 grid cells (2%) without any samples. The model output might be biased as 48% of the samples is discarded, mostly samples from dry regions. Therefore, the results from the dry regions need to be interpreted with care.



The samples that pass the forest filter (31902 samples) were used to run CLASS for the soil moisture sensitivity scenarios as these scenarios are based on the forest scenario. However, due to different model input, CLASS produced some unrealistic
output. We filtered this output using the same postprocessing filters as for the land cover scenarios (Table A2).

## 2.7 Validation

To validate the output of the CLASS model we compare different variables with the ERA5 data. We validate the output for the bare soil scenario using all grid cells that in reality have a short and tall vegetation cover that is equal to or smaller than 0.1. We validate the output for the forest scenario using all grid cells that in reality have a tall vegetation cover that is equal
to or larger than 0.8. As a result, 268 grid cells (9%) are used to validate the bare soil scenario output, and 279 grid cells (10%) are used to validate the forest scenario output. We validate the model output of the last time step (3 pm).

To validate the model output we studied the distribution of values for different output variables of CLASS and compared this to the distribution within the ERA5 data (at 3 pm). We study the distribution by calculating the mean, median and standard deviation of all samples for the relative humidity, potential temperature, boundary layer height and CAPE. These
distributions show some discrepancies between the ERA5 data and the CLASS output (Table A3). The ERA5 data has some uncertainty due to biases in its underlying observations and models, making it challenging to interpret differences between ERA5 and CLASS. These biases are larger in dry areas compared to wet areas, though spatial and temporal patterns are less uncertain than absolute values. Therefore, we focus on the spatial patterns while interpreting our results.

## 2.8 Model output interpretation

To determine the convective rainfall potential from the model output we calculated CAPE and whether the ABL and LCL cross. We calculated CAPE using the cape_sin function of the MetPy python package (https://unidata.github.io/MetPy/latest/api/generated/metpy.calc.cape_cin.html). For the input of this function, surface pressure, and the profiles of potential temperature and specific humidity of the free atmosphere were assumed constant throughout the day and were obtained from the ERA5 input data. The potential temperature and specific humidity of the
ABL and the jumps of potential temperature and specific humidity at the top of the ABL were obtained from the CLASS model output.

To analyze the sensitivity of different output variables and convective rainfall potential to land cover type we calculated the difference between both land cover scenarios for each sample specifically and the average of these differences for each grid cell. In addition, we determined the amount of samples that have sufficient CAPE as well as a crossing of the ABL and LCL.
The latter is also done for the different soil moisture scenarios. To analyze the uncertainty of the convective rainfall potential we also study the convective rainfall potential for a change in BLH, LCL and CAPE of ±10%.



## 3 Results

### 3.1 Land cover sensitivity

The differences in boundary layer characteristics between the forest and bare soil scenarios show significant spatial variation
(Fig. 2 and A4-A7). For ten different output variables, the anomalies between the forest scenario and bare soil show the same
spatial pattern (Fig. 2) that overlaps with the spatial variability of soil moisture content (Fig. A1). Hence, the impact of
forestation on several variables seems to be closely related to soil moisture content across the Mediterranean Basin. We find
two different effects of forestation, depending on soil moisture level.

For wet regions where soil moisture content is relatively large (Fig. A1), the additional energy that becomes available due to
the reduced albedo for the forest scenario, is used for evapotranspiration (latent heat flux) whereas the sensible heat flux
remains relatively consistent or decreases when forest cover increases (Fig. 2). The increase in evapotranspiration enhances
the moisture content of the ABL, enhancing the specific humidity (up to 0.002 kgkg$^{-1}$), relative humidity (up to 20%) and the
jump in specific humidity at the top of the boundary layer (up to 0.002 kgkg$^{-1}$, Fig. 2). As the sensible heat flux decreases in
the relatively wet regions, the potential temperature within the ABL is reduced as well (Fig. 2). The jump in potential
temperature at the top of the ABL shows a small increase due to the enhanced forest cover.

In these wet regions, both increases and decreases in CAPE occur as a result of forestation (up to 800 Jkg$^{-1}$, Fig. 2). CAPE
strongly increases over mountainous regions (e.g., south Turkey, west of Greece and Adriatic coastline). For the relatively
wet regions, the height of the ABL is little affected by an increase in forest cover (Fig. 2) due to the relatively small change
in the sensible heat flux. The LCL is also little affected by an increase in forest cover for most grid cells in the relatively wet
regions (Fig. 2). However, for a few grid cells the LCL decreases due to an increase in humidity.

For the relatively dry regions where the soil moisture content is relatively low (Fig. A1), the additional energy that becomes
available due to a decrease in albedo results in an increase in the sensible heat flux (up to 250 Wm$^{-2}$, Fig. 2). Compared to
the relatively wet regions, less soil moisture is available for evapotranspiration in these relatively dry regions. The higher
sensible heat flux contributes to the warming (up to 5 K) and drying (over -0.002 kgkg$^{-1}$) of the ABL (Fig. 2), as well as the
deepening of the ABL (up to 1 km). The deepening of the ABL resulting from an increase in forest cover enhances the
entrainment of warm air from the free atmosphere into the ABL (Miralles et al., 2014). This entrainment contributes to the
high potential temperatures over forest compared to over bare soil for dry regions and brings dry air from the free
atmosphere into the ABL. As the latent heat flux does is not affected by forestation in these relatively dry regions, the ABL
becomes less humid. The warming and drying of the ABL contribute to the rising of the LCL (more than 5 km).
Furthermore, over these relatively dry regions, the land cover change does not affect CAPE much, which shows that for the
Mediterranean Basin, first, the presence of sufficient soil moisture seems to be necessary for CAPE to develop and, second,
that an increase in temperature alone does not necessarily enhance CAPE.

There is a spatial variation in the impact of land cover change on convective rainfall potential across the Mediterranean
Basin (Fig. 3). Despite that the ABL height and LCL vary little over the relatively wet regions there is an increase in the



number of crossings when the forest cover increases (Fig. 3). Hence, only a small change in the ABL height and/or LCL
       seems to have a significant impact on the convective rainfall potential. We find that 8.2% of the samples have convective
       rainfall potential when the grid cells are covered in forest and 4.9% of the samples have a rainfall potential when the grid
       cells are covered in bare soil (Fig. 3). Focusing on the two conditions for convective rainfall potential, for the bare soil
       scenario, 10% of the samples have a crossing and 16% of the samples have sufficient CAPE ($\geq 400$ Jkg$^{-1}$) and for the forest
scenario, 26% of the samples have a crossing and 32% of the samples have sufficient CAPE (Fig. 3).

       For the western Iberian Peninsula, Italy, the Adriatic coastline and western Turkey, most of the grid cells show an increase in
       convective rainfall potential when the forest cover is enhanced (Fig. 3). However, approximately thirty of the grid cells (1%)
       show a decrease in convective rainfall potential when going from bare soil to forest (Fig. 3). For the relatively dry regions
       the increase in forest cover enhances the LCL, which explains why there is no convective rainfall potential here (Fig. 2). To
test the robustness of the results we varied the ABL height, LCL, and CAPE with 10%. Both for a 10% increase as well as a
       10% decrease of these variables, the spatial variability of the sensitivity of the convective rainfall potential to land cover type
       is little affected. However, the total amount of samples with convective rainfall potential is affected (Fig. A8-A10).



**Figure 2: Difference between forest scenario and bare soil scenario for different output variables of CLASS. Each grid shows the mean value of all samples for that specific grid cell. The total amount of samples per grid varies between 0 and 20 due to the postprocessing step. The output variables that are shown are the latent heat flux (LE), sensible heat flux (H), specific humidity (q), relative humidity (RH), jump in specific humidity at the top of the boundary layer (dq), potential temperature (theta), jump in potential temperature at the top of the boundary layer (dtheta), convective available potential energy (CAPE), boundary layer height (BLH), and lifting condensation level (LCL).**




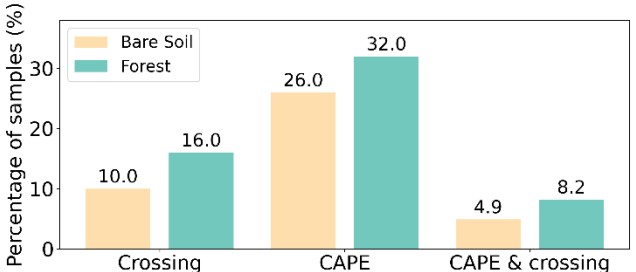

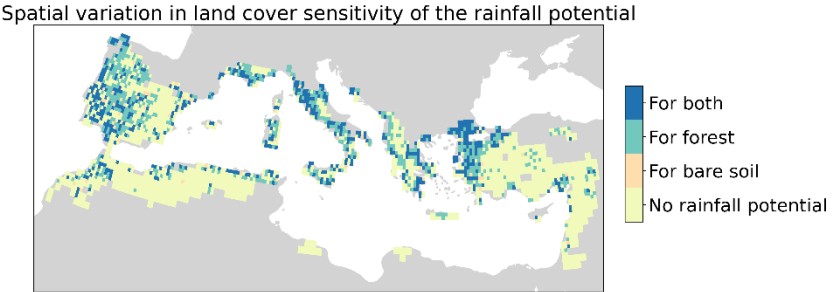

**Figure 3: Top: The percentage of samples with a crossing, sufficient CAPE (≥ 400 Jkg⁻¹), or both sufficient CAPE and a crossing for the Mediterranean Basin when covered in either forest or bare soil. Bottom: The spatial variability of the land cover sensitivity of the convective rainfall potential, i.e., there is both a crossing and sufficient CAPE. This plot indicates for each grid cell if most** 255 **samples have a convective rainfall potential over bare soil, forest or both land cover types or if they have no convective rainfall potential in any of the samples.**

## 3.2 Soil moisture sensitivity

The model output for the different soil moisture scenarios shows that the fraction of samples with a crossing of the ABL and LCL at 3 pm increases linearly with the soil moisture content (Fig. 4). This relationship is determined for the soil moisture 260 content ranging between the wilting point and field capacity. In addition, the fraction of samples for which CAPE exceeds 400 Jkg⁻¹ increases linearly with the soil moisture content (Fig. 4), supporting our previous finding that CAPE increases over relative wet regions when forest cover increases. With a change in soil moisture, the rate of change in the number of simulations with sufficient CAPE is larger (i.e., steeper slope) compared to the rate of change in the number of crossings. Finally, the fraction of samples with convective rainfall potential also increases linearly with soil moisture content. With a 265 change in soil moisture the convective rainfall potential has a similar rate of change as the fraction of samples with a crossing (Fig. 4).

The convective rainfall potential does not increase everywhere in the Mediterranean basin when soil moisture increases (Fig. 4). In some grid cells in the central parts of the Iberian peninsula, Italy, Turkey, the Balkans, and northern Africa convective rainfall potential is more pronounced when soil moisture equals wilting point compared to when soil moisture equals field 270 capacity (Fig. 4). This could be explained by a decrease in the sensible heat flux when soil moisture increases, minimizing





the ABL development and crossing. However, a larger amount of grid cells show that the convective rainfall potential increases with an increase in soil moisture content.

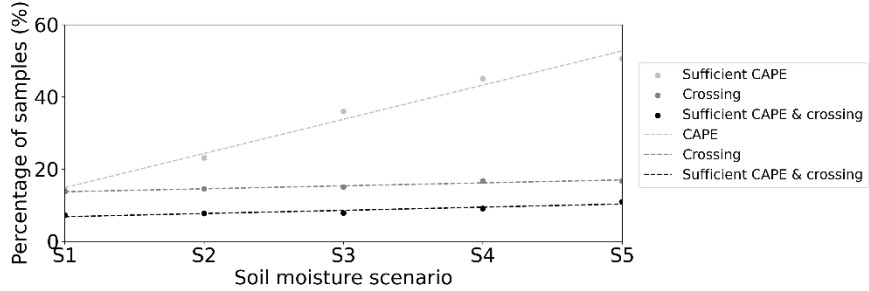

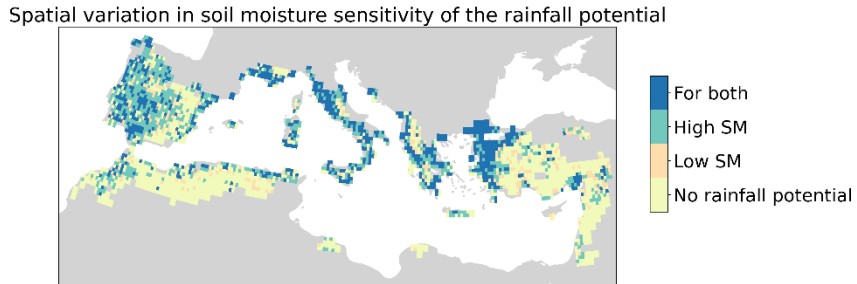

**Figure 4: Top: The percentage of samples with a crossing, sufficient CAPE (≥ 400 Jkg⁻¹), or both sufficient CAPE and a crossing for varying soil moisture content. The different soil moisture scenarios increase linearly from wilting point (S1, 0.15 m3m-3) to field capacity (S5, 0.35 m3m-3). The dotted lines are the regression lines (CAPE: R2 = 0.98, crossing: R2 = 0.93, CAPE & crossing: R2 = 0.87). Bottom: The spatial variability of the soil moisture sensitivity of the convective rainfall potential, i.e., a crossing of the ABL and LCL and sufficient CAPE. This plot indicates for each grid cell if most samples have a convective rainfall potential for soil moisture scenario 1, soil moisture scenario 5, or both soil moisture scenarios, or no convective rainfall potential in any of the samples.**

## 4 Discussion

### 4.1 Boundary layer development under change

Changing the land cover in the Mediterranean Basin from bare soil to forest affects the atmospheric boundary layer (ABL) characteristics and the convective rainfall potential locally. There is a rainfall potential when the ABL reaches the lifting condensation level (LCL) and when there is sufficient convective available potential energy (≥ 400 Jkg⁻¹ , CAPE), parcels can rise over a depth great enough to trigger precipitation. There is correlation between LCL and CAPE, as both are computed following a rising parcel. Therefore, changes that lead to an earlier LCL crossing may also result in an increase in CAPE (Yin et al., 2015). We find that the impact of forestation on convective rainfall potential varies with soil moisture and that it does not increase the convective rainfall potential everywhere in the Mediterranean Basin. A change in land cover from bare soil to forest enhances convective rainfall potential mainly in relatively wet regions. However, not all grid cells in





the relatively wet regions have an increase in convective rainfall potential after forestation and some grid cells, specifically in the center of the Iberian Peninsula, show a decrease in convective rainfall potential after forestation. In dry regions there is no convective rainfall potential for either the bare soil scenario or the forest scenario.

The soil moisture content also has an impact on whether an increase in forest cover has a cooling or warming effect. For 295 relatively dry regions, the increased net radiation is mainly transferred into sensible heat, contributing to the warming of the ABL. For relatively wet regions, the increase in net radiation enhances evapotranspiration, preventing the potential temperature within the ABL to rise and even contributing to cooling in some locations. This cooling effect can explain why some locations have a convective rainfall potential in the bare soil scenario and not in the forest scenario as it minimizes the deepening of the ABL.

When soil moisture is varied across the Mediterranean Basin we find that, overall, the convective rainfall potential increases linearly with soil moisture content. However, for some grid cells across the basin, the convective rainfall potential decreases with increasing soil moisture content. These grid cells have a convective rainfall potential when soil moisture equals wilting point, yet these cells do not have a convective rainfall potential when the soil moisture equals field capacity. This negative relation between soil moisture and convective rainfall potential could be explained by a stronger cooling effect over wetter 305 soils. Additionally, it shows that the relationship between soil moisture and convective rainfall potential is more complex than the overall linear relationship that we found.

An increase in soil moisture does not necessarily relate linearly to evapotranspiration but their coupling is dependent on the aridity of the region and is stronger in relatively dry regions (Seneviratne et al., 2010). However, this stronger coupling in dry regions does not necessarily reflect in an increase in rainfall as only a limited range of free atmospheric conditions 310 allows for convective rainfall potential (Findell and Eltahir, 2003b, a; Juang et al., 2007; Konings et al., 2011). This may explain why the effect of an increase in soil moisture on convective rainfall potential seems to be most pronounced close to regions that already have a convective rainfall potential for less soil moisture. Here the atmospheric conditions are already near the threshold for convective rainfall potential.

In the eastern and southern Mediterranean Basin, atmospheric conditions appear to limit convective rainfall potential, as 315 increasing soil moisture, even up to field capacity, does not effectively trigger precipitation (Findell and Eltahir, 2003a). This regional difference may be attributed to the large-scale atmospheric circulation patterns characteristic of the Mediterranean Basin, which create contrasting weather conditions between its eastern and western regions (Roberts et al., 2012). This climatic 'see-saw' likely explains why enhanced soil moisture supports convective rainfall potential in the western Mediterranean but fails to do so in the eastern part. In the eastern Mediterranean, unfavorable free atmospheric conditions 320 may inhibit the development of convective rainfall potential despite increased soil moisture.

## 4.2 Discussion on the simulation

The results presented here need to be interpreted carefully for several reasons. First, the CLASS model provides a simple description of the processes within the ABL and therefore, this study is limited to studying convective triggering, as it is



assumed that forestation is applied locally and that the upper atmosphere remains unchanged because of it. Consequently, we
approximated the potential for convective precipitation using CAPE and the crossing of the ABL and LCL. Furthermore, as
clouds are not explicitly modelled, there is some uncertainty in the energy balance. Clouds increase the albedo as they
partially reflect incoming radiation, which is not included in the model. This feedback would likely have a cooling effect
(Cerasoli et al., 2021; Fraedrich and Kleidon, 1999), specifically over wet regions, which has a negative impact on the
development of the ABL and CAPE (Seeley and Romps, 2015; Vilà-Guerau de Arellano et al., 2015).
Second, as the model describes the ABL as a 1-dimensional column, the horizontal spatial component is not taken into
account. This horizontal spatial component is essential to simulate the impact of surface roughness on convection (Pielke,
2001) or to simulate the development of a sea breeze during the day. Hence, moisture convergence, which contributes to the
development of convective precipitation, may be underestimated in the CLASS model. The spatial component can be
included in more complex models such as a large eddy simulation model. Although CLASS does not simulate all processes it
allows us to improve our understanding of the main interactions between the land surface and the ABL that are responsible
for convective precipitation.

Finally, ERA5 has some uncertainty, specifically over dry regions (https://confluence.ecmwf.int/display/CKB/ERA5,
accessed 16-12-2024). However, by using ERA5 as input data we can represent the present-day climate realistically for a
large region. To the best of our knowledge, this is the first study to use ERA5 as input data for CLASS. Previous studies
mainly use in-situ observations as input for and validation of an ABL model (Vilà-Guerau de Arellano et al., 2004; Wouters
et al., 2019), due to the size of our study region this was not possible. Compared to the uncertainty of exact values over dry
regions, the spatial and temporal variation of the ERA5 data have a higher accuracy. Therefore, we mainly focus on the
spatial patters. Our results showed that a change in BLH, LCL or CAPE of 10% does not affect the spatial pattern of the land
cover sensitivity of the convective rainfall potential (Fig. A8-A10), so the inaccuracy of the exact values may be of less
importance.

## 4.3 The uncertainty of rainfall potential under change

In line with previous literature, our results indicate that forestation can have both cooling and warming effects in the
Mediterranean Basin. Ruijsch et al. (2024) found that land restoration has a net cooling effect in parts of northern Africa due
to increased evaporation, whereas Portmann et al. (2022) observed a net warming effect throughout the entire Mediterranean
Basin due to global forestation. King et al. (2024) did not find a significant temperature effect of forestation in temperate
regions. While these studies highlighted the potential for both warming and cooling effects from forestation, they did not
acknowledge the dependence of these effects on soil moisture content. The coupling between vegetation and soil moisture
has been previously established (Bonan, 2008; Materia et al., 2022) and is especially strong in arid regions (Forzieri et al.,
2017).
The increase in precipitation potential due to forestation that we found is also in line with previous climate modelling efforts
that found an increase of terrestrial precipitation due to global forestation (Fraedrich and Kleidon, 1999; Gibbard et al., 2005;





Portmann et al., 2022). This precipitation increase ranges from 0.8% (Portmann et al., 2022) to 100% (Fraedrich and Kleidon, 1999). Although these previous studies suggest an increase in precipitation due to forestation, the impact on local precipitation remains unclear because the local and remote effects are not isolated. There are some modelling efforts that were able to identify local reductions in precipitation resulting from deforestation (Luo et al., 2022; Winckler et al., 2017) as well as a data analysis effort that was able to identify a positive effect of vegetation on local water availability and precipitation (Cui et al., 2022). Our study supports these results that suggest that forestation may enhance local precipitation in addition to terrestrial precipitation.

Similar to previous studies, our results indicate a positive relation between CAPE and soil moisture content (Emanuel, 2023; Leutwyler et al., 2021; Liu et al., 2022). Even though we found that CAPE increases linearly with soil moisture, it is expected that this relationship is more complex. This is expected because first, convective storms might not develop over too wet soils due to a larger amount of energy that is needed to raise an air parcel from the surface to the level of free convection (Emanuel, 2023) and second, storms could also intensify when moving towards drier areas(Liu et al., 2022). Nevertheless, soil moisture availability seems to be an important factor to establish CAPE and convective rainfall potential, especially over mountainous regions (Liu et al., 2022), and as a result, precipitation is promoted over wet soils and reduced over dry soils during summer, indicating a soil moisture-rainfall feedback (Ardilouze et al., 2022; Findell and Eltahir, 2003b; Leutwyler et al., 2021).

By only modelling the local processes under current climate conditions, remote effects on convective rainfall potential as well as the impact of atmospheric warming are overlooked. Changes in temperature and humidity due to forestation will likely affect the transport of moisture and heat (Lian et al., 2022; Meier et al., 2021; Pielke, 2001; Portmann et al., 2022; Staal et al., 2024b), and therefore, remote precipitation events. However, the exact impact remains unclear and the warming of the atmosphere and sea surface due to climate change may have a bigger impact on precipitation (King et al., 2024), enhancing uncertainty. Nevertheless, for the Mediterranean Basin, a large fraction of evaporated water recycles within the region (Batibeniz et al., 2020; Schicker et al., 2010; Theeuwen et al., 2024). Therefore, it is expected that an increase in evaporation due to forestation will likely affect precipitation elsewhere within the Mediterranean Basin. However, whether this would result in regional wetting or drying remains unclear. Studies in other regions show that restoration efforts can be beneficial for rainfall within the wider region (Tian et al., 2022).

## 4.4 Regreening to enhance the convective rainfall potential locally

The results of this study indicate where forestation may contribute to more rainfall. However, it should be noted that due to competing land use (e.g., agriculture or cities), soil type, or climatic conditions some of these areas are not suited for forestation efforts (Noce et al., 2017; Tóth et al., 2008). Nevertheless, for forestation efforts to contribute to convective rainfall, they need to be conducted in relatively wet regions as it seems that forestation makes wet regions wetter and dry regions drier. Not only sufficient soil moisture needs to be available, also a relatively humid atmosphere is beneficial to enhance rainfall locally (Theeuwen et al., 2023, 2024). Evidently, the soil moisture content and atmospheric humidity are





closely linked (Seneviratne et al., 2010), specifically in regions that are neither extremely wet, nor extremely dry (Konings et al., 2011). Moving towards the south, the wetness of the Mediterranean Basin decreases. This suggests that it is more likely that forestation enhances convective rainfall in the north than in the south of the Mediterranean Basin.

Forestation efforts need to be conducted in regions where the conditions of the free atmosphere allow for the development of convective rainfall potential. The climatic 'see-saw' that has been observed in the mediterranean Basin (Roberts et al., 2012)

may explain why the atmospheric conditions in the western and northern parts of the basin seem to be more favorable to trigger convective rainfall potential through regreening than the conditions in the east and south. Furthermore, forestation mainly seems to increase convective rainfall potential in grid cells adjacent to those already exhibiting significant convective rainfall potential over bare soil, likely because the atmospheric conditions in these areas are favorable for convective rainfall potential

Specifically coastal regions show potential for convective rainfall enhancement through forestation. This is observed in Italy, Greece and northern Africa and corresponds to previous research efforts that show a negative relation between local evaporation recycling and the distance to the nearest coast (Theeuwen et al., 2024). Finally, mountains seem to promote the crossing of the ABL and LCL (e.g., in the South of Turkey). Complex terrain enhances the sensitivity of convection to soil moisture (Liu et al., 2022). Therefore, our results suggest that forestation in coastal regions with elevated terrain and

relatively high soil moisture may contribute to local rainfall, specifically in the northern and western part of the basin.

## 5 Conclusions

The goal of this study was to investigate the impact of forestation on the convective rainfall potential in the Mediterranean Basin during early summer. There is convective rainfall potential when the atmospheric boundary layer (ABL) and lifting condensation level (LCL) cross and the convective available potential energy (CAPE) is at least 400 Jkg$^{-1}$. Using the CLASS

model, the boundary layer development was simulated for two land cover scenarios: each grid cell covered fully in (1) bare soil, and (2) forest. We found that forestation enhances CAPE, the amount of crossings, and the convective rainfall potential in various locations within the Mediterranean Basin. The impact of forestation on the development of the ABL varies across the Mediterranean Basin and relates to soil moisture. In relatively dry regions, forestation contributes to the warming and drying of the ABL, resulting in a deepening of the ABL and increase of the LCL, which overall does not contribute to a

crossing. In relatively wet regions, forestation moistens the ABL and enhances CAPE, which overall contributes to convective rainfall potential. However, for a few relatively wet grid cells forestation seems to reduce the convective rainfall potential. Furthermore, additional simulations of the Mediterranean Basin with different soil moisture scenarios ranging from wilting point to field capacity underlined the crucial role of soil moisture for convective rainfall potential. Similar to the impact of forestation on convective rainfall potential, for a few grid cells an increase in soil moisture has a negative impact

on the convective rainfall potential suggesting that there is an optimal amount of soil moisture. We also found that soil moisture is not the only driver of the crossing. It seems that for specific atmospheric conditions no crossing occurs, even



when soil moisture content reaches field capacity. Nonetheless, our results suggest that, overall, there is a positive relation between soil moisture content and the convective rainfall potential. Thus to potentially enhance local rainfall through forestation, forestation initiatives in the Mediterranean Basin could be conducted in relatively wet regions close to the coast 425 and over elevated terrain.



## Appendix A

**Table A1: List of input variables for the CLASS model. The data inputs are divided into parameters and variables, from which the latter group is divided into variables that can directly be obtained from ERA5 data and variables that needed additional calculation steps. Constants are obtained from ECMWF IFS documentation (ECMWF, 2010).**


| Variable | | Dataset |
|---|---|---|
| Initial surface pressure | | ERA5 at single level |
| Initial surface temperature (skin temperature) | | ERA5 at single level |
| **Variable** | **Variable calculated with** | **Dataset** |
| Initial mixed layer potential temperature | Temperature at 2 pressure levels (800 and 750 hpa) | ERA5 at pressure levels |
| Initial mixed layer specific humidity | Specific humidity 2 pressure levels (800 and 750 hpa) | ERA5 at pressure levels |
| Initial temperature jump | Temperature at 2 pressure levels (800 and 750 hpa) | ERA5 at pressure levels |
| Initial specific humidity jump | Specific humidity at 2 pressure levels (800 and 750 hpa) | ERA5 at pressure levels |
| Advection of heat (sum of day) | Temperature and wind speed at 100 m neighboring cells | ERA5 at single level |
| Advection of moisture (sum of day) | Specific humidity and wind speed at 100 m neighboring cells | ERA5 at single level |
| Lapse rate potential temperature | Temperature at 2 pressure levels (800 and 750 hpa) | ERA5 at pressure levels |
| Lapse rate specific humidity | Specific humidity at 2 pressure levels (800 and 750 hpa) | ERA5 at pressure levels |
| Volumetric water content (case 1 and 2) | Water content top 3 soil layers | ERA5 at single level |
| Temperature soil | Temperature top 3 soil layers | ERA5 at single level |
| **Parameter** | **value** | **Source** |
| Initial boundary layer height | 100 m | Assumption |
| Roughness length heat | 0.0013 m (bare soil) & 2 m (forest) | ECMWF IFS documentation (Cy36r1, Table 11.4) |
| Roughness length momentum | 0.013 m (bare soil) & 2 m (forest) | ECMWF IFS documentation (Cy36r1, Table 11.4) |



| Vegetation fraction | 0 (bare soil) & 1 (forest) | Decided by experiment |
|---|---|---|
| LAI (bare) | 0.01(bare soil) & 5 (forest) | ECMWF IFS documentation (Cy36r1, Table 8.1) |
| Albedo | 0.25 (light colored bare soil) & 0.15 (forest) | ECMWF IFS documentation (Figure 11.17 and Figure 11.18) |
| Wilting point (WP) | 0.15 | ECMWF IFS documentation (Cy36r1, Table 8.8) |
| Saturation soil | 0.44 | ECMWF IFS documentation (Cy36r1, Table 8.8) |
| Field capacity (FC) | 0.35 | ECMWF IFS documentation (Cy36r1, Table 8.8) |
| Soil moisture (soil moisture scenarios) | [ WP – FC] | - |
| Minimum resistance transpiration (forest) | 250/50 (mixed forest) | ECMWF IFS documentation (Cy36r1, Table 8.1) |
| Minimum resistance soil evaporation (bare soil) | 50 s/m | ECMWF IFS documentation (chapter 8 page 111) |
| Aerodynamic resistance | 0 (bare soil) & 0.03 (forest) | ECMWF IFS documentation (Cy36r1, Table 8.2) |

**Table A2: List of postprocessing filters that were applied to filter out the unrealistic output of CLASS.**

| Postprocessing filters |
|---|
| Soil moisture $\leq 1$ ($m^3m^{-3}$) |
| $0 \leq$ Relative humidity $\leq 10$ (-) |
| $100 \leq$ boundary layer height $\leq 4000$ (m) |
| Potential temperature $\leq 323$ (K) |
| $0 \leq$ Jump potential temperature $\leq 3$ (K) |
| $-0.008 \leq$ Jump specific humidity $\leq 0$ ($kgkg^{-1}$) |





**Table A3: The mean, median, and standard deviation of different variables for the bare soil validation samples and the forest validation samples. For temperature we take the potential temperature from CLASS and the temperature at 2 m from ERA5. Following an ideal temperature profile, in the surface layer of the ABL the potential temperature is lower than in the mixed layer. It is not possible to make a comparison between bare soil and forest as these values are obtained for different locations.**

|  |  | Bare Soil | | Forest | |
| --- | --- | --- | --- | --- | --- |
|  |  | CLASS | ERA5 | CLASS | ERA5 |
| **BLH** | Mean | 2608 | 774 | 2112 | 1728 |
|  | Median | 2626 | 570 | 2010 | 1651 |
|  | St. dev. | 387 | 651 | 652 | 777 |
| **Temp.** | Mean | 30 | 23 | 30 | 25 |
|  | Median | 29 | 22 | 30 | 25 |
|  | St. dev. | 6 | 5 | 6 | 6 |
| **RH** | Mean | 0.74 | 0.88 | 0.67 | 0.80 |
|  | Median | 0.73 | 0.91 | 0.64 | 0.80 |
|  | St. dev. | 0.37 | 0.08 | 0.29 | 0.08 |
| **CAPE** | Mean | 117 | 197 | 596 | 103 |
|  | Median | 0 | 2.5 | 79 | 0 |
|  | St. dev. | 341 | 431 | 1054 | 276 |

**Table A4: Minimum value of specific variables for samples with rainfall potential. If a single value is indicated there is only a lower 'threshold'. If a range of values is indicated this variable has a lower and upper 'threshold'. The upper 'threshold' is the maximum value of all samples with rainfall potential. Besides the minimum and maximum we also show the 5th percentile and the 95th percentile.**

|  | Bare soil | | Forest | |
| --- | --- | --- | --- | --- |
|  | Threshold | 5% threshold | Threshold | 5% threshold |
| **BLH** (m) | 1141 | 1543 | 1279 | 1773 |
| **theta** (K) | $288 - 312$ | 294-305 | $290 - 312$ | $295 - 307$ |
| **dtheta** (K) | $0.2 - 2.4$ | $0.8 - 2.0$ | $0.3 - 3.0$ | $1.0 - 2.7$ |
| **q** (kgkg$^{-1}$) | 0.004 | 0.006 | 0.004 | 0.005 |
| **dq** (kgkg$^{-1}$) | -0.007 | -0.006 | -0.008 | -0.007 |



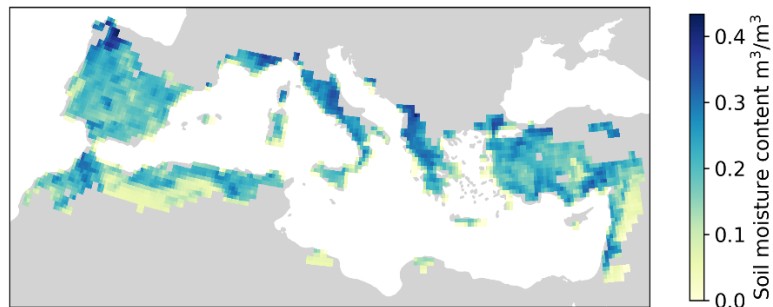

**Figure A1: The soil moisture content in the top layer for all regions around the Mediterranean Sea that have a Mediterranean climate according to the Köppen climate classification. This plot shows the mean soil moisture content in May and June for the years 2013-2022.**

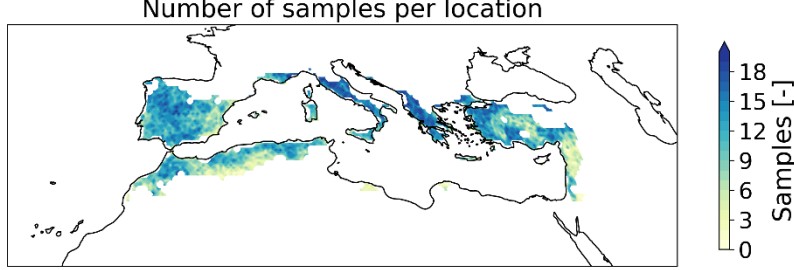

**Figure A2. (a) The spatial distribution of the number of samples that pass through the filter for each grid cell in the Mediterranean Basin.**





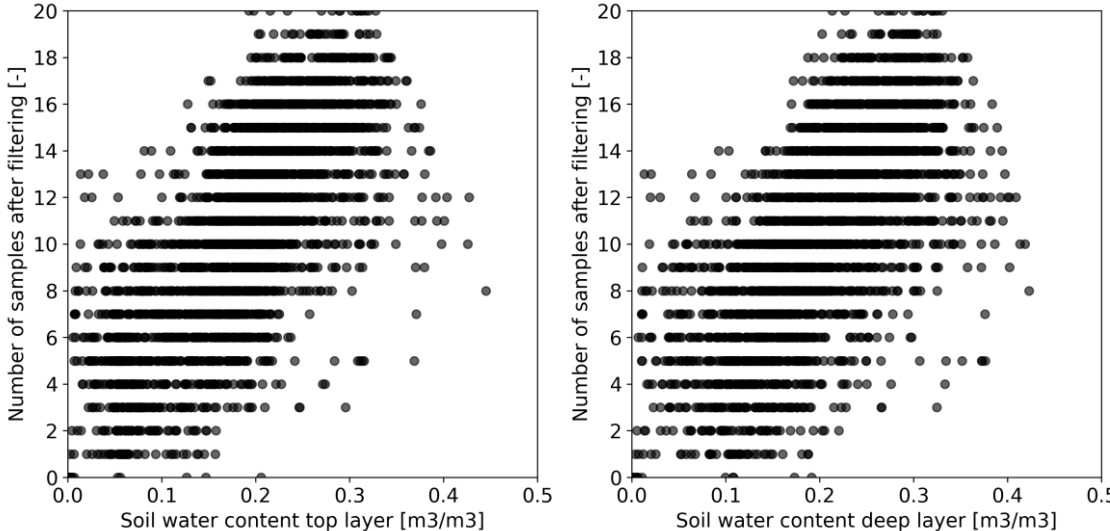

**Figure A3: The relationship between the mean soil water content for each grid cell and the number of samples for each grid cell after filtering for unrealistic output of CLASS. Soil water content in the top layer (left) and soil water content in the deep layer (right). Each scatter point represents one grid cell.**








**Figure A4: Output of the bare soil scenario and forest scenario for all samples that pass the postprocessing filter for boundary layer height (BLH), lifting condensation level (LCL), convective available potential energy (CAPE), relative humidity (RH), potential temperature (Theta), and the jump in potential temperature at the top of the boundary layer (dTheta). The orange line indicates the identity line (x=y).**




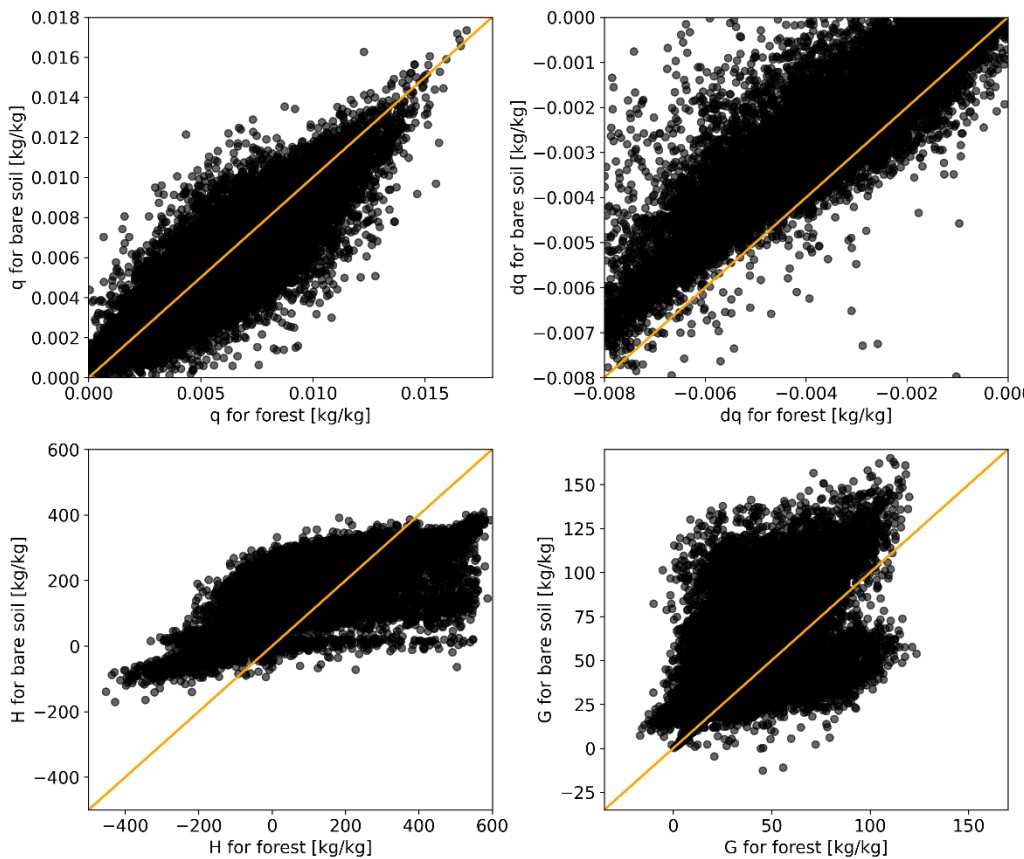

**Figure A5: Output of the bare soil scenario and forest scenario for all samples that pass the postprocessing filter at 3 PM for specific humidity (q), jump in specific humidity at the top of the boundary layer (dq), sensible heat flux (H), ground heat flux (G). The orange line indicates the identity line (x=y).**




**Figure A6: CLASS output for the bare soil scenario. For each variable the mean value per grid cell is shown. Note that due to filtering the amount of samples varies among the grid cells. The output that is shown is the latent heat flux (LE), sensible heat flux (H), specific humidity (q), relative humidity (RH), jump in specific humidity at the top of the boundary layer (dq), potential temperature (theta), jump in potential temperature at the top of the boundary layer (dtheta), convective available potential energy (CAPE), boundary layer height (BLH), and lifting condensation level (LCL).**




**Figure A7: CLASS output for the forest scenario. For each variable the mean value per grid cell is shown. Note that due to filtering the amount of samples varies among the grid cells. The output that is shown is the latent heat flux (LE), sensible heat flux (H), specific humidity (q), relative humidity (RH), jump in specific humidity at the top of the boundary layer (dq), potential temperature (theta), jump in potential temperature at the top of the boundary layer (dtheta), convective available potential energy (CAPE), boundary layer height (BLH), and lifting condensation level (LCL).**






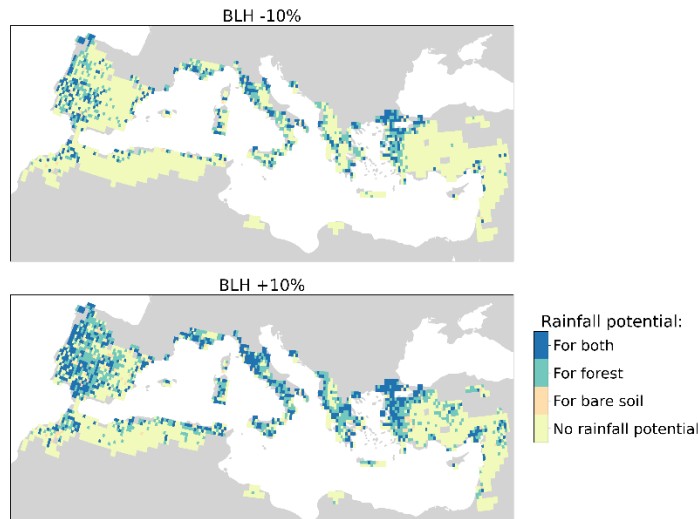

**Figure A8: Sensitivity of rainfall potential to a change in the boundary layer height (BLH). Top: a change in BLH of -10%,**
**bottom: a change in BLH of +10%. This plot indicates for each grid cell if most samples have a rainfall potential over bare soil,**
**forest, both land cover types or no rainfall potential in any of the samples.**

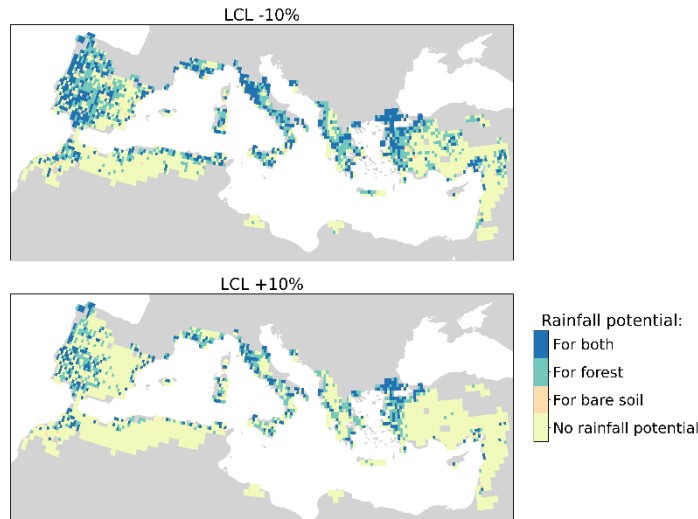

**Figure A9: Sensitivity of rainfall potential to a change in the lifting condensation level (LCL). Top: a change in LCL of -10%,**
**bottom: a change in LCL of +10%. This plot indicates for each grid cell if most samples have a rainfall potential over bare soil,**
**forest, both land cover types or no rainfall potential in any of the samples.**



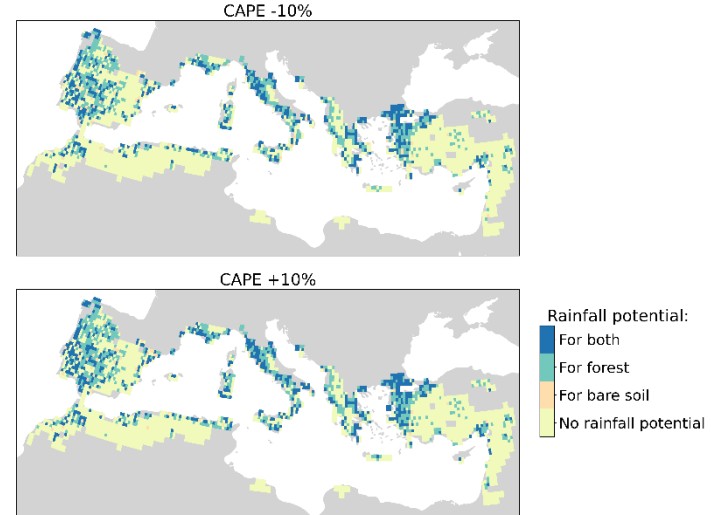

**Figure A10: Sensitivity of rainfall potential to a change in the convective available potential energy (CAPE). Top: a change in CAPE of -10%, bottom: a change in CAPE of +10%. This plot indicates for each grid cell if most samples have a rainfall potential**
**over bare soil, forest, both land cover types or no rainfall potential in any of the samples.**

**Appendix B: Calculations for input variables**

The following equations show how the potential temperature, lapse rate of the free atmosphere, the temperature jump at the top of the boundary layer and the advection of heat were calculated. Correspondingly, the specific humidity gradient of the free atmosphere, the specific humidity jump at the top of the boundary layer and the advection of moisture were calculated.


**Potential temperature:**

$$\theta = T(\frac{P_0}{P})^{R/C_p} \qquad (A1)$$

The potential temperature is derived from the temperature ($T$) and the corresponding pressure ($P$). $C_P$ is the specific heat capacity and $R$ is the gas constant of air. $R/C_p$=0.286. $P_0$ is the standard reference pressure: 1013.25 hPa.


**Lapse rate:**

$$\theta_{FA}(z) = \theta_{FA}(0) + \gamma_\theta z \qquad (A2)$$

The lapse rate ($\gamma_\theta$) is calculated using the potential temperature at two pressure levels (750 and 800 hPa). Using Equation A2 we derive:

$$\gamma_\theta = \frac{d\theta}{dz} = \frac{d\theta}{dP}\frac{dP}{dz} = \frac{\theta_{750}-\theta_{800}}{P_{750}-P_{800}} \cdot -\rho g \qquad (A3)$$

Here, $\rho$ equals the density of dry air (1.225 kg m$^{-3}$) and $g$ the gravitational constant (9.81 ms$^{-2}$).





**Initial temperature jump at top boundary layer:**

The height of the ABL ($h_0$) is set to 100 m. The temperature or the mixed ABL is assumed to be:

$$\theta_{ABL} = \frac{\theta_{FA}(h_0) + \theta_{FA}(0)}{2} = \theta_{FA}(z = 50m) \tag{A4}$$

The temperature at the top of the boundary layer equals $\theta_{FA}(h_0)$ and therefore the temperature jump at the top of the boundary layer is equal to:

$$\Delta\theta = \theta_{FA}(z = 50m) - \theta_{FA}(h_0) = \gamma_\theta \Delta z \tag{A5}$$

In which $\Delta z$ equals 50 m.

**Advection of heat:**

To calculate the fluxes between grid cells, the following equation is used:

$$Q = \rho C_p \frac{\partial uT}{\partial x} + \rho C_p \frac{\partial vT}{\partial y} \tag{A6}$$

$$\frac{\partial uT}{\partial x} = \frac{1}{r_{earth}\cos(lat)} \frac{\partial uT}{\partial \lambda} \tag{A7}$$

$$\frac{\partial vT}{\partial y} = \frac{1}{r_{earth}} \frac{\partial vT}{\partial \vartheta} \tag{A8}$$

Here, u is the zonal wind speed, $v$ is the meridional wind speed, $\lambda$ is the longitude, $\theta$ is the latitude, and $r_{earth}$ is the radius of the earth. We assume constant temperature and velocity with height within our boundary layer. We use the wind velocity at 100 m.

**Code availability**

The code for CLASS is available through http://classmodel.github.io/. Code that can be used to analyse the data will be made available on GitHub before publication of the manuscript.

**Data availability**

We will make the output from the CLASS model available on Zenodo before publication of the manuscript. The ERA5 data is freely available at the Copernicus Climate Change Service (C3S) Climate Data Store https://cds.climate.copernicus.eu (Hersbach et al., 2023). The shapefile of the study region is obtained from (Theeuwen et al., 2024)

**Author contribution**

JJET designed the study with contributions from all authors, conducted the preprocessing and postprocessing steps, carried out the analyses, and wrote the first draft of the manuscript. SW conducted the model simulations, interpreted the output of



CLASS, and was involved in the writing of the first draft of the manuscript. AS was involved in writing the first draft of the

manuscript. All authors contributed to the discussion and the final version of the manuscript.

**Competing interests**

The authors declare that they have no conflict of interest.

**Acknowledgements**

This work was performed in the cooperation framework of Wetsus, European Centre of Excellence for Sustainable Water

Technology (www.wetsus.eu). Wetsus is co-funded by the Dutch Ministry of Economic Affairs and Climate Policy, the

Northern Netherlands Provinces and the Province of Fryslan. The authors would like to thank the participants of the natural

water production theme (DEME and The Weather Makers) for the fruitful discussions and financial support. AS

acknowledges support from the Talent Programme grant VI.Veni.202.170 by the Dutch Research Council (NWO).

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
