# Peer review of "Forest favours conditions for convective precipitation in the Mediterranean Basin"

_EGUsphere, 2025_

## Author Comment (AC1)

**Response to reviewer comment from reviewer #1 for manuscript egusphere-2025-289**

In this initial response, we aim to briefly address the main issues raised by the reviewer to facilitate a prompt exchange and encourage discussion. Our focus is on the key concerns to ensure an efficient and constructive dialogue. If the discussion concludes positively and we are invited to revise our manuscript based on the reviewers' feedback, we will then provide a detailed response addressing all the points raised in the rebuttal.

> This study simulates the impact of land cover change and soil moisture availability on boundary layer development in the Mediterranean Basin using the Chemistry Land-surface Atmosphere Soil Slab (CLASS) model. By comparing CAPE and ABL height across different land cover and soil moisture scenarios, the authors determine that convective rainfall potential increases when vegetation fraction is increased over wet regions and increases linearly with soil moisture content. While the results of the experiment and its design are interesting and compelling contributions to the land-atmosphere interactions literature, I have some major reservations about the framing of the study around "forestation" as a potential climate mitigation strategy. I recommend that the manuscript undergo major revisions to reevaluate and clarify its research goals and interpretations.

We are thankful that the reviewer took the time to provide us with constructive feedback that will help to improve the manuscript. We also thank the reviewer for expressing this study is an interesting and compelling contribution to the land-atmosphere interactions literature.

**Framing and language**

> Motivation: The abstract motivates the study by first identifying the Mediterranean Basin as a "climate change hotspot" that may be "prone to future drying." The authors then follow up this statement by noting that "Previous studies indicate the effect of forests on precipitation remains unclear for the Mediterranean Basin" before diving into a description of the study. The link between climate change, future drying, and vegetation-precipitation coupling is not clear at all from these two sentences. The introduction does marginally better in explaining "forestation may increase freshwater availability" and "forestation... may enhance rainfall." What's missing is the underlying implication that either A) we expect forestation to occur in this region or B) forestation is being considered as a climate mitigation strategy. If you decide to stick with the forestation angle, please explain and expand on this instead of leaving the readers to try to connect the dots. Assuming B based on the discussion of regreening later in the paper, how realistic is this strategy and how seriously is it being considered for the region? How confident are we that the Mediterranean will experience drying given that hydrological trends tend to experience large disagreement between models? Some important context is missing.

We agree with the reviewer that the motivation could be clarified both in the abstract and introduction and we thank the reviewer for this comment. We will clarify the motivation for this study and include that forestation is being considered as a climate mitigation strategy. We will clarify that the hydrological effects of forestation should not be overlooked as they may have positive or negative effects on local freshwater availability. With our study we aim to get a better understanding of where forest may have positive hydrological effects (i.e., an increase in convective rainfall potential), and where is may have negative hydrological effects (i.e., a decrease in rainfall potential and expected drying due to enhanced evapotranspiration).

Furthermore, the reviewer raises several questions about the confidence of the projected drying in Mediterranean regions and whether regreening is seriously considered for this region. First, while we agree that the projections of climate models have uncertainty, it is highlighted by the IPCC that ongoing warming has been observed in the Mediterranean Basin already affecting the ecosystems here (Ali et al., 2022). Furthermore, with high confidence it has been estimated that warming in this Mediterranean region has exceeded global average rates as well as that temperature extremes and heatwaves have increased in intensity, number, and length, particularly during summer (Ali et al., 2022). We will add this reference to the discussion. Second, while covering the entire Mediterranean Basin in forest is highly unrealistic and undesirable, there are projects that aim to restore forests in the Mediterranean Basin (e.g., https://www.decadeonrestoration.org/restoring-mediterranean-forests, https://www.unep.org/news-and-stories/story/fighting-fire-forests-across-mediterranean). Additionally, research shows that forest management (including afforestation) can improve carbon sequestration to cope with climate change (Ruiz-Peinado et al., 2017). We will include this information and these references in our manuscript. We believe that this will improve the motivation for our research, and therefore, we would like to thank the reviewer for their comment.

> **Defining and Interpreting "Forestation":** It is not clear to me whether the study actually addresses the question of how forestation would affect rainfall in the region. "Forestation" and other terms like "regreening" and "restoration" that are used liberally in the paper are 1) not well defined and 2) typically imply that there is a gradual increase in vegetation cover over some timescale that takes into account the planting and growing process. Instead, what this study does is answer the question of "how convective rainfall potential would be different over the Mediterranean Basin if the region were covered in forest" by dramatically altering the land cover properties of the grid cells across different model runs (i.e., a sensitivity study). There are no dynamical considerations in the experiment setup, so I am not confident the results can be interpreted as the climate response to "an increase in forest cover," at least not in the context of any real-world replanting strategy. In other parts of the paper, the authors describe the results using phrases like "The differences in boundary layer characteristics between the forest and bare soil scenarios show significant spatial variation," which is much more accurate (L199). While these results certainly have important implications for the use of forestation as a climate mitigation strategy, the title and language misrepresent the study's scope. Perhaps I am misunderstanding what "forestation" means to the authors, but I think the study would fit much better in a vegetation-precipitation/land-atmosphere coupling context rather than with the current climate change angle.

The reviewer points out that "forestation", "regreening", and "restoration" are used liberally in the paper and that they are not well defined. We thank the reviewer for pointing this out and agree that we should clarify these concepts when revising the manuscript. In addition, the reviewer correctly states that forestation results in a gradual increase in vegetation cover and not in a sudden increase in vegetation cover as was modelled in this study. We agree that during the development of a forest the land atmosphere interactions will vary over time. To fully understand the impact of a specific land use change, this gradual development should be considered as well as other effects, such as ecological succession, which are also not included in this study as this is beyond the scope of this manuscript. The aim of our manuscript is to get a better understanding of where forest may have positive hydrological effects (i.e., an increase in rainfall potential), and where it may have negative hydrological effects (i.e., a decrease in rainfall potential and expected drying due to enhanced evapotranspiration). We believe that the results

of this study show, once a  mature forest has grown, the regions in the Mediterranean Basin that may contribute to more rain locally. We will highlight the implication of our results better in the manuscript. We do want to note that for the regions where a mature forest may contribute to more rain, a more detailed study is necessary. Such an analysis could include the gradual development of a forest.

In summary, we agree with the reviewer that the term "forestation" may contribute to the expectation that this study models the maturing of a forest. To address this issue we will first, more clearly state the aim. Second, to prevent the misuse of the term forestation we will introduce the term "forestation" in the introduction when we provide context of the study, and include it in the discussion to discuss the implications of our results. Throughout the manuscript we will use more accurate phrases such as, L199 as pointed out by the reviewer, and the term "mature forest". Finally, we will reconsider the use of "forestation" in the title of this manuscript.

**Science clarifications**

**Sampling:** 10 years seems like an insufficient length of time to establish a climatology for the region (L83). The description of the sampling method in 2.4 is extremely unclear to me. What does it mean to run the model 20 times for each grid cell with two random days being sampled for each year? Is the study only simulating the atmospheric conditions during 20 random days over the 10-year time period? This, in addition to the high number of samples that had to be filtered out, is very concerning. Please also specify that CLASS is a single column/grid cell model.

We thank the reviewer for pointing out that the sampling method is not described clearly in our manuscript. We will make sure to clarify the sampling method in the manuscript. To answer the question from the reviewer about the number of runs per 10-year period, for each grid cell we run the model 20 times. This results in a total of 57360 samples.

In this manuscript we identify spatial patterns in land-atmosphere interactions for the Mediterranean Basin. As we were mainly interested in spatial patters, we decided to have a full spatial coverage of the study region. To get statistically significant results we decided to select 20 samples for each location. We agree with the reviewer that filtering out a large number of samples raises uncertainty, specifically in some parts in the south of the study region where soil moisture is low. However, more samples do not improve the statistical significance of our analysis as for the relatively dry regions, the same percentage of samples to pass/not pass the filtering step. This would result in a similar uncertainty. We would also like to stress that the results of this study do not provide an actual prediction of what may happen to convective rainfall over a mature forest, the results give an indication of the impact of soil moisture and land cover on convective precipitation. We will discuss the implications of filtering out a large amount of the samples in more detail both in the methods, and discussion sections.

Furthermore, a longer climatology has both advantages and disadvantages. Within a 30-year period, the warming trend would be more pronounced, which may offset the results. We will clarify our motivation and implications for a 10-year study period in the manuscript.

Finally, in the methodology we will specify that CLASS is a single column model. We thank the reviewer for pointing this out.

**CAPE Analysis:** Why was a threshold of 400 J/kg chosen for the CAPE analysis? Given that the authors calculate CAPE using the metpy cape_cin (typo in L186) function, CIN should also be included in the analysis and would provide a more robust standard for

determining the likelihood of convective initiation. Recent studies (Emanuel, 2023; Zhang et al., 2023) have also shown that the development of high CIN over wet soils is essential to explaining the development of high CAPE in both models and in observations. Given that the relationship between CAPE and soil moisture is one of the study's main results, the discussion in L364-373 could be expanded and some of that literature should be mentioned earlier in the introduction (L64).

We thank the reviewer for their helpful suggestion. If we are invited to submit a revised version of our manuscript we will include CIN in our analysis. We will discuss in more detail how CIN will be included in the rebuttal letter.

**Spatial Correlation with Soil Moisture Regimes:** Given that the interpretation relies heavily on an understanding of the different wet and dry locations in the basin, Figure A1 should be included in the main paper to be more easily accessible. The spatial variations in Figure 2 for CAPE are quite scattered compared to the more coherent patterns in BLH and LCL. Could the authors perform some sort of correlation between the ABL/CAPE changes and the soil moisture variations across the region? That is, how well do the ABL/CAPE changes map onto the soil moisture regimes? Also, given that the study is currently framed in the context of future climate change, please comment on how we expect those regimes to change in the future. If moisture decreases in the region, will these results still hold? Please also discuss how realistic an average ΔLCL of over 8km is and rescale the plot in Figure 2.

We thank the reviewer for this suggestion. We will include Figure A1 in the main text of this manuscript. Additionally, we will study correlations between soil moisture and differences in ABL and CAPE for the two different land cover types. This will provide insight into how the variations in ABL and CAPE due to different land cover types overlap with the different soil moisture regimes. Finally, we will discuss in more detail how the future climate may affect the different outcomes for bare soil and forest.

**P-E and Moisture Recycling:** Given that the study's goal is to understand "future drying" and the changes in precipitation potential are accompanied by changes in evapotranspiration, there should be some considerations of P-E in the discussion of "wet gets wetter" (Abstract).

This would be an interesting addition and we thank the reviewer for thinking along. However, as this method does not allow to quantify the change in precipitation, it is not possible to study how the different land cover types affect P-E across the region. We agree with the reviewer that this information would be highly valuable to better understand whether a forest would contribute to drying or wetting. To account for this issue, we will more carefully phrase the subsentence "wet gets wetter". We will take a close look at the manuscript to see if we should make similar changes in other parts of the manuscript as well.

In addition, we would like to clarify the goal of this study. The goal is not to study future drying specifically. The goal is to study the impact of land cover on the convective rainfall potential to asses where in the Mediterranean Basin a mature forest may contribute to more rainfall locally.

Finally, we would like to thank the reviewer once more for their valuable feedback.

**References:**

Ali, E., W. Cramer, J. Carnicer, E. Georgopoulou, N.J.M. Hilmi, G. Le Cozannet, and P. Lionello, 2022: Cross-Chapter Paper 4: Mediterranean Region. In: Climate Change 2022: Impacts, Adaptation and Vulnerability. Contribution of Working Group II to the Sixth Assessment Report of the Intergovernmental Panel on Climate Change [H.-O. Pörtner, D.C. Roberts, M. Tignor, E.S. Poloczanska, K. Mintenbeck, A. Alegría, M. Craig, S. Langsdorf, S. Löschke, V. Möller, A. Okem, B. Rama (eds.)]. Cambridge University Press, Cambridge, UK and New York, NY, USA, pp. 2233–2272, doi:10.1017/9781009325844.021.

Ruiz-Peinado, R., Bravo-Oviedo, A., López-Senespleda, E., Bravo, F., & Del Río, M. (2017). Forest management and carbon sequestration in the Mediterranean region: A review. *Forest Systems*, *26*(2), eR04S. https://doi.org/10.5424/fs/2017262-11205

---

## Author Comment (AC2)

**Author's response to reviewer comment from reviewer #2 for manuscript egusphere-2025-289**

In this initial response, we aim to briefly address the main issues raised by the reviewer to facilitate a prompt exchange and encourage discussion. Our focus is on the key concerns to ensure an efficient and constructive dialogue. If the discussion concludes positively and we are invited to revise our manuscript based on the reviewers' feedback, we will then provide a detailed response addressing all the points raised in the rebuttal.

> The study examines the influence of forest cover on convective precipitation potential in the Mediterranean Basin, using the CLASS model to simulate the atmospheric boundary layer (ABL) response to changes in land cover (bare soil vs. forest) and soil moisture. While the paper is well-organized and offers valuable insights into land-atmosphere interactions, there are some major reservations I have regarding the methodology, underlying assumptions, interpretation of results, and the clarity of key definitions that should be addressed and revised in the manuscript before publication.

We are happy to hear that the reviewer believes our manuscript offers valuable insights into land-atmosphere interactions and that they find our manuscript well structured. We are thankful the reviewer took their time to provide us with valuable feedback that will help us to improve the manuscript.

**1. Definition of key terms**

> • The terms "forestation," "regreening," and "land restoration" are used throughout the manuscript but lack precise definitions. Please clarify what these terms mean in the context of this paper.

We thank the reviewer for pointing this out, we will carefully reconsider the use of those terms. The terms that we will use will be clarified in the introduction of the manuscript. Finally, we will be more consistent our terminology. Reviewer #1 gave similar feedback, indicating the importance to properly address this point.

> • **L285:** The term "parcels" is introduced for the first time in the Discussion section without prior definition, though it is an important concept in the calculation of CAPE. To ensure clarity, the authors should define this term earlier in the manuscript, preferably in the Methods section, and briefly explain its relevance to the study's atmospheric processes.

The reviewer raises a very good point. In the methods section we will describe in more detail how CAPE is calculated, which offers the opportunity to introduce the term "parcels".

> • **L55 and L186:** Given the study's focus on CAPE and since CAPE is a calculated quantity rather than a height like the ABL or LCL, the authors should explicitly provide the equation used for its computation in the main text. Simply referencing the MetPy Python function is insufficient, as it does not clarify the exact formulation or assumptions applied in this study. Including the full equation and associated assumptions will improve transparency and reproducibility.

We thank the reviewer for this suggestion. We will include the calculation, full equations and their assumptions of CAPE in more detail in the methods section of our manuscript. This also allows us to properly introduce the term "parcels" in the methods section, as raised by the previous comment.

**2. Study setup**

- **L57: "When both the ABL and LCL cross and the CAPE is at least 400 Jkg-1 there is convective rainfall potential."** The phrasing "both the ABL and LCL cross" needs clarification: does this mean the ABL height exceeds the LCL? Additionally, while CAPE represents atmospheric instability, a threshold of 400 J/kg does not inherently indicate precipitation without considering other key factors such as CIN, mid-tropospheric moisture, and large-scale forcing (Emanuel, 2023). Can the authors clarify the reasoning behind this threshold and account for additional necessary conditions for convective rainfall?

We thank the reviewer for pointing out how we could improve our study set-up. We will more clearly explain the definition of a crossing between the LCL and ABL in our manuscript. To clarify the definition in this discussion, the ABL and LCL cross when ABL ≥ LCL. Hence, there is also a crossing when the LCL and ABL have the same height.

Based on boundary layer dynamic theory and model studies, we came to the conclusion that the parameterization of ABL>=LCL and CAPE>400 J/kg gives a good representative of potential convective precipitation. We will clarify this better in the method section. However, taking into account the comments of both reviewers we believe it is important to add convective inhibition (CIN) in our analysis as well. CIN is the amount of energy (in J/kg) that prevents an air parcel from rising freely through the atmosphere to reach the level where convection starts. CIN halts convection unless overcome by sufficient forcing. In addition to studying the relation between land use type and CAPE, we will also study the relation between land use type and CIN.

Finally, we will include a paragraph in our discussion that describes what other factors (e.g., mid-tropospheric moisture, and more) affect convective precipitation to put the results of our study better into perspective. We would like to note that our set-up of the CLASS model includes an advection term. Therefore, the large-scale forcing is accounted for in our model. We will describe this more clearly in our manuscript.

- **L47: "To isolate the local effects of changes in land cover on local precipitation, a different model approach is necessary."** Please clarify what is meant by "a different model approach." Specifically, elaborate on the key differences between this approach and existing methods, and justify why the proposed model is better suited for isolating local effects on precipitation.

What we refer to with "a different model approach" is using a model that does only simulate the local processes that are affected by land use changes. In climate models that use land cover scenarios, a change in land cover upwind from location X may affect precipitation, or other variables, in location X. This makes it challenging to isolate how a land cover change in location X itself may affect the local processes. We will explain this more clearly in line 47 and use this to motivate our model selection. We thank the reviewer for pointing how we can improve the clarity of our manuscript.

- **L83: "This simulation is done for early summer as this is the start of the dry season."** It is not clear to me why the early summer (May and June) time period is significant for study when it is stated in L93 that there is late spring and summer convective precipitation in the region. Have the authors considered expanding the period of study (which may also improve the study sampling rate)? Please elaborate on this.

During spring/early summer the coupling between the land surface and the atmosphere is stronger compared to other seasons (Ardilouze et al., 2022; Benetó & Khodayar, 2023; Gates & Liess 2001; Lombardo & Bitting 2024). This implies that vegetation relates most to convective precipitation during this time of year. We will add this motivation in the method section. We will also support this statement with the following literature:

"Local evaporation seems to be the most important moisture source during the dry season (10 days integrated contribution is 310.70 10 12 mm yr 1 km 2)." (Gómez-Hernández et al., 2013, p. 6787)

"When comparing Tables 3a and 3b, it is seen that the role of local evaporative processes is more relevant during the dry season." (Gómez-Hernández et al., 2013, p. 6789)

"As mentioned in section 1, large interior extensions present arid or semiarid characteristics and attain the maximum of precipitation in spring, making the recycling contribution essential to describe the precipitation regime locally. On the contrary, the lack of a clear link between recycling and precipitation from October to February confirms that land-atmosphere mechanisms do not play a relevant role on rainfall in the winter half of the year." (Rios-Entenza et al., 2014, p. 5905)

"In Iberia, both conditions (that is, a synoptic configuration favoring convection and sufficient soil moisture availability) reach their maximum coupling in spring and early summer and can be effectively assessed in early spring (March)." (Rios-Entenza et al., 2014, p. 5909)

- **L93: "Although precipitation falls predominantly in winter, during late spring and summer there is convective precipitation in the region."** Given that the study relies on historical data from ERA5, it would be beneficial to include a figure illustrating which grid cells recorded observed precipitation and which are currently bare soil vs. forest. This would help assess the spatial distribution of convective precipitation and clarify how well the modeled convective rainfall potential corresponds to observed precipitation events given the current state of the region. This would be relevant to strengthen the study's conclusions and recommendation on the need for forestation in different areas to "potentially enhance local rainfall through forestation". Can the authors provide such a figure to support this statement?

We thank the reviewer for this suggestion, we believe that such a figure to be a valuable addition to our manuscript and we will include it in the appendix.

- **Section 2.6 Postprocessing.** I am concerned about the disproportionate exclusion of dry regions compared to wet regions due to the filtering process. The bias introduced by the increased sampling size in wetter regions undermines the generalizability of the authors' conclusion that both forestation and an increase in soil moisture can contribute to convective rainfall potential. Given that nearly half of the samples—primarily from dry regions—were excluded, can the authors clarify how this bias affects their findings? Additionally, how do the authors justify applying these conclusions across the Mediterranean Basin area of study when dry regions are underrepresented in the results? (particularly for Fig. 3). Can the authors comment on this?

We agree with the reviewer that our conclusion should be formulated more carefully and that we should explain the implications of the filtering process more clearly. We will discuss the resulting bias in more detail in the discussion section of our manuscript. We will more carefully

phrase the conclusion and discuss their uncertainty for the drier regions in the Mediterranean Basin taking the bias into account.

To summarize the bias here shortly: The results have a higher uncertainty in the south of the Mediterranean Basin due to the smaller amount of samples that pass the filter. However, within these relatively dry regions there are quite some grid cells that have 10 (50%) or more samples passing the filtering step (Figure A2). Still, there are some regions where our results have a relatively high uncertainty, i.e., in Libya, Lebanon, and Syria. The results in these regions should be interpreted very carefully. We will carefully review the manuscript to determine whether it is necessary to include statements to inform the reader that a careful interpretation is necessary. In addition, we will discuss the uncertainty of our results and its variability across the study region in more detail in the manuscript.

- **Section 2.7 Validation.** The authors mention some numbers to classify "short and tall vegetation cover", but it is not clear to me how these values are retrieved or calculated. Please clarify the source and methodology used to define these classifications.

We thank the reviewer for pointing out this is not clear. These data are obtained from the ERA5 dataset. In the ERA5 dataset tall vegetation is considered as: evergreen trees, deciduous trees, mixed forest/woodland, and interrupted forest; Short vegetation is considered as: crops and mixed farming, irrigated crops, short grass, tall grass, tundra, semidesert, bogs and marshes, evergreen shrubs, deciduous shrubs, and water and land mixtures. Not all these vegetation types are found in the study region. We will include these details in section 2.7 of our manuscript.

- **Section 2.8 Model output interpretation**, **L195: "To analyze the uncertainty of the convective rainfall potential we also study the convective rainfall potential for a change in BLH, LCL and CAPE of ±10%."** The authors state that they analyze the uncertainty of convective rainfall potential by varying BLH, LCL, and CAPE by ±10%. However, it is unclear why these 10% variations are chosen and if there is any statistical significance towards the conclusion that the "inaccuracy of the exact values may be of less importance" in L344.

These 10% variations are chosen to get more insight into the robustness of the spatial patterns we identified. We were interested in the sensitivity of the model output to small variations in BLH, LCL, and CAPE. However, we were not interested in whether this sensitivity is linear or non-linear. Therefore, we decided to study the implications of 10% variations. The decision to study 10% variations is somewhat arbitrary, yet it suffices to analyze the relative sensitivity of the model output to the three variables. The statement in line 344 is based on the uncertainty of the ERA5 dataset. The exact values in ERA5 are subject to uncertainty as mentioned in the official documentation of ERA5 (https://confluence.ecmwf.int/display/CKB/ERA5%3A+data+documentation). However, the spatial resolution in the ERA5 data has a smaller uncertainty. The uncertainty in the ERA5 data was the main motivation to study the uncertainty in the spatial patterns. We selected variations of 10% to better understand if a change in either BLH, LCL, or CAPE, due to uncertainty in the exact values of ERA5 would have effects on the spatial patterns that were obtained in our study and whether these patterns are robust despite small variations in BLH, LCL and CAPE.

- It also seems to me that this uncertainty is related to the "inaccuracy of the exact values" and assesses the sensitivity of results to minor perturbations in key variables. How do the authors account for the impact of sampling bias and

dataset exclusions (particularly in dry regions) on the robustness of their conclusions (particularly for Fig. 3 and 4 and associated discussion)?

We thank the reviewer for pointing this out. We believe that we can improve the discussion on the impact of the filtering step on the results in Fig. 3 and Fig. 4. We will describe the impact of the filtering step in more detail in the discussion. In this discussion we will refer to how the results presented in Figs. 2-4 may be affected by this filtering step.

- **For all figures referencing the rainfall potential color scale.** Please include in the caption how rainfall potential is defined. Specifically, what constitutes a grid cell to "have a convective rainfall potential" or "have no convective rainfall potential" in the sample?

We thank the reviewer for pointing this out. We will include the definition of rainfall potential in the captions of the figure and describe the difference between "a convective rainfall potential" and "no convective rainfall potential".

**3. General manuscript proofreading**

- Throughout the manuscript, there are multiple instances of missing commas and periods, which affect general readability and clarity. I recommend a thorough grammatical review to improve sentence structure, punctuation, and overall flow. In particular, some sentences lack necessary commas for readability, and certain sections contain run-on sentences that would benefit from clearer punctuation. See L27, L47, L48 for some (not all) examples. Also, see L186: "cape_sin" should be corrected to the correct function name "cape_cin", and L394: "mediterranean" should be capitalized. A careful proofreading by the authors would enhance the manuscript's clarity.

We thank the reviewer for pointing out that the readability and clarity of the manuscript could be improved overall. We will make sure to conduct a final review by multiple authors in which we pay close attention to the grammar. We thank the reviewer for already pointing out some of the lines which we should revise.

Finally, we would like to thank the reviewer for taking the time to provide valuable feedback on our submitted manuscript.

**References:**

Ardilouze, C.; Materia, S.; Batté, L.; Benassi, M.; Prodhomme, C. Precipitation Response to Extreme Soil Moisture Conditions over the Mediterranean. *Clim Dyn* **2022**, *58* (7), 1927–1942. https://doi.org/10.1007/s00382-020-05519-5.

Benetó, P.; Khodayar, S. On the Need for Improved Knowledge on the Regional-to-Local Precipitation Variability in Eastern Spain under Climate Change. *Atmospheric Research* **2023**, *290*, 106795. https://doi.org/10.1016/j.atmosres.2023.106795.

Dümenil Gates, L.; Liess, S. Impacts of Deforestation and Afforestation in the Mediterranean Region as Simulated by the MPI Atmospheric GCM. *Global and Planetary Change* **2001**, *30* (3–4), 309–328. https://doi.org/10.1016/S0921-8181(00)00091-6.

Gómez-Hernández, M.; Drumond, A.; Gimeno, L.; Garcia-Herrera, R. Variability of Moisture Sources in the Mediterranean Region during the Period 1980–2000. *Water Resources Research* **2013**, *49* (10), 6781–6794. https://doi.org/10.1002/wrcr.20538.

Lombardo, K.; Bitting, M. A Climatology of Convective Precipitation over Europe. *Monthly Weather Review* **2024**, *152* (7), 1555–1585. https://doi.org/10.1175/MWR-D-23-0156.1.

Rios-Entenza, A.; Soares, P. M. M.; Trigo, R. M.; Cardoso, R. M.; Miguez-Macho, G. Moisture Recycling in the Iberian Peninsula from a Regional Climate Simulation: Spatiotemporal Analysis and Impact on the Precipitation Regime. *Journal of Geophysical Research: Atmospheres* **2014**, *119* (10), 5895–5912. https://doi.org/10.1002/2013JD021274.

---

## Author Response (AR1)

Subject: Rebuttal for manuscript egusphere-2025-289

Dear Andrew Feldman,

Thank you for giving us the opportunity to revise our manuscript with the original title "Forestation tends to create favorable conditions for convective precipitation in the Mediterranean Basin". We are grateful for all the time and work the reviewer put into their constructive and valuable feedback. Their comments helped to improve our manuscript.

Based on the reviewers' comments we made some adjustments to the methodology. First, we included convective inhibition (CIN) in our analysis, as suggested by the reviewers. Second, we study the correlation between soil moisture and boundary layer height, lifting condensation level, convective available potential energy, and CIN, to better understand how soil moisture relates to the convective rainfall potential in the forest cover and bare soil scenarios.

Furthermore, we revised the text throughout the manuscript to improve the framing and shift the focus from the impact of forestation on convective rainfall potential to the impact of forests. Additionally, we better explain some of the key concepts of our study and we revised the discussion and conclusion to more clearly highlight the uncertainty due to the filtering step. Finally, multiple authors conducted a final grammar check to improve the overall readability of the manuscript. Below we explain in more detail how we implemented each of the reviewers' comments.

We hope that the revisions and clarifications we have made in response to the reviewers' comments have sufficiently addressed their concerns and improved the overall quality of the manuscript. We believe that the revised version is now suitable for publication in *Biogeosciences* and we look forward to your evaluation.

On behalf of all authors,

Kind regards,

Jolanda Theeuwen

**Detailed response to reviewer comment from reviewer #1**

The reviewer's comments are presented in blue and the authors' response is presented in black. The line numbers refer to the line numbers in the document that includes all the tracked changes.

This study simulates the impact of land cover change and soil moisture availability on boundary layer development in the Mediterranean Basin using the Chemistry Landsurface Atmosphere Soil Slab (CLASS) model. By comparing CAPE and ABL height across different land cover and soil moisture scenarios, the authors determine that convective rainfall potential increases when vegetation fraction is increased over wet regions and increases linearly with soil moisture content. While the results of the experiment and its design are interesting and compelling contributions to the land-atmosphere interactions literature, I have some major reservations about the framing of the study around "forestation" as a potential climate mitigation strategy. I recommend that the manuscript undergo major revisions to reevaluate and clarify its research goals and interpretations.

We are thankful that the reviewer took the time to provide us with constructive feedback that will help to improve the manuscript. We also thank the reviewer for expressing this study is an interesting and compelling contribution to the land-atmosphere interactions literature. Below, for each of the points raised by the reviewer, we discuss in detail which changes were made to the manuscript.

**Framing and language**

Motivation: The abstract motivates the study by first identifying the Mediterranean Basin as a "climate change hotspot" that may be "prone to future drying." The authors then follow up this statement by noting that "Previous studies indicate the effect of forests on precipitation remains unclear for the Mediterranean Basin" before diving into a description of the study. The link between climate change, future drying, and vegetation-precipitation coupling is not clear at all from these two sentences.

The introduction does marginally better in explaining "forestation may increase freshwater availability" and "forestation... may enhance rainfall." What's missing is the underlying implication that either A) we expect forestation to occur in this region or B) forestation is being considered as a climate mitigation strategy. If you decide to stick with the forestation angle, please explain and expand on this instead of leaving the readers to try to connect the dots. Assuming B based on the discussion of regreening later in the paper, how realistic is this strategy and how seriously is it being considered for the region? How confident are we that the Mediterranean will experience drying given that hydrological trends tend to experience large disagreement between models? Some important context is missing.

We agree with the reviewer that the motivation could be clarified both in the abstract and the introduction, and we thank the reviewer for this valuable comment. To better highlight the motivation of our study, we included the following lines in the abstract of our revised manuscript: "Through carbon sequestration, forests may mitigate climate change and reduce future drying. Nevertheless, the effect of forests on freshwater availability in the Mediterranean Basin is uncertain. Trees contribute to enhanced evapotranspiration, which may enhance drying; the resulting impact on precipitation in the Mediterranean Basin, however, remains unclear."

Furthermore, the reviewer raises important questions regarding the confidence in projected drying in Mediterranean regions and whether regreening is seriously being considered for this area. First, while we acknowledge that climate model projections involve uncertainties, the IPCC highlights that ongoing warming has already been observed in the Mediterranean Basin, affecting local ecosystems (Ali et al., 2022). Moreover, it has been estimated with high confidence that warming in the Mediterranean region has exceeded global average rates, and that temperature extremes and heatwaves have increased in intensity, frequency, and duration, particularly during summer (Ali et al., 2022). Second, while covering the entire Mediterranean Basin in forest is highly unrealistic and undesirable, there are indeed projects aiming to restore forests in the region (e.g., https://www.decadeonrestoration.org/restoring-mediterranean-forests, https://www.unep.org/news-and-stories/story/fighting-fire-forests-across-mediterranean). Additionally, research shows that forest management, including afforestation, can enhance carbon sequestration and contribute to climate change mitigation (Ruiz-Peinado et al., 2017).

To further clarify the motivation of our study, we included the following lines in the introduction (lines 34–42 and 46): "Previous research estimated with high confidence that warming in the

Mediterranean Basin has exceeded global average rates and temperature extremes and heatwaves have increased in intensity, number, and length, particularly during summer (Ali et al., 2022). Forestation initiatives that contribute to enhanced forest cover, such as forest restoration, afforestation, forest management and more, are carried out across the globe to mitigate climate change and reverse land degradation (United Nations Environment Programme, 2021). Such initiatives are also realized in the Mediterranean Basin, where forest management can contribute to carbon sequestration (Ruiz-Peinado et al., 2017). In addition to affecting the climate through carbon sequestration, forests may increase freshwater availability when the increase in evapotranspiration promotes precipitation (Cui et al., 2022)."... "However, increased evapotranspiration can also reduce streamflow (Galleguillos et al., 2021) and therefore, (Staal et al., 2024b) some forests may contribute to local drying."

By including this text, the revised version of our manuscript now reflects that forestation is being considered as a climate mitigation strategy in the Mediterranean Basin. In addition, we clarify that the hydrological effects of forestation for climate mitigation should not be overlooked, as they may have both positive and negative effects on local freshwater availability. Finally, we address the confidence in climate change projections. We believe these changes help clarify the relevance of our research, and we thank the reviewer once again for their helpful comment.

**Defining and Interpreting "Forestation":** It is not clear to me whether the study actually addresses the question of how forestation would affect rainfall in the region. "Forestation" and other terms like "regreening" and "restoration" that are used liberally in the paper are 1) not well defined and 2) typically imply that there is a gradual increase in vegetation cover over some timescale that takes into account the planting and growing process. Instead, what this study does is answer the question of "how convective rainfall potential would be different over the Mediterranean Basin if the region were covered in forest" by dramatically altering the land cover properties of the grid cells across different model runs (i.e., a sensitivity study). There are no dynamical considerations in the experiment setup, so I am not confident the results can be interpreted as the climate response to "an increase in forest cover," at least not in the context of any real-world replanting strategy. In other parts of the paper, the authors describe the results using phrases like "The differences in boundary layer characteristics between the forest and bare soil scenarios show significant spatial variation," which is much more accurate (L199). While these results certainly have important implications for the use of forestation as a climate mitigation strategy, the title and language misrepresent the study's scope. Perhaps I am misunderstanding what "forestation" means to the authors, but I think the study would fit much better in a vegetation-precipitation/landatmosphere coupling context rather than with the current climate change angle.

The reviewer points out that the terms "forestation," "regreening," and "restoration" are used liberally in the paper and are not well defined. We thank the reviewer for this observation and agree that these concepts should be clarified. To ensure greater consistency, we decided to use only the term "forestation." We define this term in the following sentence: "Forestation initiatives that contribute to enhanced forest cover, such as forest restoration, afforestation, forest management and more, are carried out across the globe to mitigate climate change and reverse land degradation (United Nations Environment Programme, 2021)."

In addition, the reviewer correctly states that forestation results in a gradual increase in vegetation cover and not in a sudden increase in vegetation cover as was modelled in this study. We agree that during the development of a forest the land atmosphere interactions will vary over time. To fully understand the impact of a specific land use change, this gradual development

should be considered as well as other effects, such as ecological succession, which are also not included in this study as this is beyond the scope of this manuscript. The aim of our manuscript is to get a better understanding of where forest may have a positive hydrological effect (i.e., an increase in rainfall potential). We believe that the results of this study show, once a mature forest has grown, the regions in the Mediterranean Basin that may contribute to more rain locally.

In addition, the reviewer correctly notes that forestation typically results in a gradual increase in vegetation cover, whereas our study models a sudden change. We agree that land–atmosphere interactions evolve over time during forest development. To fully understand the impact of a specific land use change, it is important to consider this gradual progression, including ecological succession. However, modeling such processes is beyond the scope of this study. The aim of our manuscript is to better understand where forests may have a positive hydrological effect—namely, an increase in rainfall potential. We believe our results show which regions in the Mediterranean Basin may contribute to increased local rainfall once a mature forest has developed.

As we agree with the reviewer that our study does not account for the gradual nature of forestation, we have adjusted the focus of the manuscript accordingly. The revised manuscript now places greater emphasis on *forests* rather than on *forestation*. We made the following changes:

- 1. **Title**: We changed the title of the manuscript to "Forest favours conditions for convective precipitation in the Mediterranean Basin."
- 2. **Terminology**: We revised the text throughout the manuscript to shift focus from forestation to forests. For instance, forestation is only briefly introduced in the introduction (line 36) to establish the relevance of the study. The rest of the introduction refers to forests rather than forestation.
- 3. **Methods and Results**: We now focus exclusively on comparing the bare soil and forest scenarios, without discussing forestation or land cover changes.
- 4. **Discussion**: We mainly focus on differences in boundary layer development and convective rainfall potential over bare soil and forest, without linking these results directly to forestation. In the final subsection of the discussion, where we consider the broader implications of our findings, we explain how our results could inform forestation initiatives. Here, we also acknowledge that a more comprehensive understanding of forestation impacts would require modeling the gradual development of forests (lines 548–552).
- 5. **Study Aim**: We rephrased the aim of the study (lines 94–95) as follows: "To assess where in the Mediterranean Basin a mature forest may contribute to more rain locally, our study compares the occurrence of convective rainfall potential over bare soil and forest."

We believe that these changes improved the quality of our manuscript and therefore, we are very grateful for this constructive comment.

**Science clarifications**

**Sampling:** 10 years seems like an insufficient length of time to establish a climatology for the region (L83). The description of the sampling method in 2.4 is extremely unclear to me.

What does it mean to run the model 20 times for each grid cell with two random days being sampled for each year? Is the study only simulating the atmospheric conditions during 20 random days over the 10-year time period? This, in addition to the high number of samples that had to be filtered out, is very concerning. Please also specify that CLASS is a single column/grid cell model.

We thank the reviewer for pointing out that the sampling method is not described clearly in our manuscript. To answer the question from the reviewer about the number of runs per 10-year period, for each grid cell we run the model 20 times. This results in a total of 57,360 samples. In the revised version of our manuscript we clearly state that we conduct 20 runs per grid cell and that this results in 57,360 samples.

In this manuscript we identify spatial patterns in land-atmosphere interactions for the Mediterranean Basin. As we were mainly interested in spatial patters, we decided to have a full spatial coverage of the study region. To get statistically significant results we decided to select 20 samples for each location. We agree with the reviewer that filtering out a large number of samples raises uncertainty, specifically in some parts in the south of the study region where soil moisture is low. However, more samples are not likely to improve the statistical significance of our analysis for the relatively dry regions as it is expected that the same percentage of samples will pass the filtering step. We would also like to stress that the results of this study do not provide an actual prediction of what may happen to convective rainfall over a mature forest, the results give an indication of the impact of soil moisture and land cover on convective precipitation.

Furthermore, a longer climatology has both advantages and disadvantages. Within a 30-year period, the warming trend would be more pronounced, which may offset the results. We will clarify our motivation for a 10-year study period in the manuscript.

To address these different points that were raised by the reviewer we included the following lines in our methodology (lines 174-180): "As the main aim of this study is to understand where in the Mediterranean Basin a forest may contribute to local rainfall we are mainly interested in spatial patterns in ABL development, and therefore, sample over the entire study region. The entire region is divided in 2868 grid cells of 0.25° × 0.25°. We decided to analyze the rainfall potential during the months May and June over a 10-year period (2013-2022). To obtain statistically significant results we conduct 20 runs per grid cell. To equally divide these runs over the 10-year study period we randomly select two days for each year. To prevent an off-set in the model output due to a warming trend we study a 10-year period rather than a 30-year period."

To address the comment about the uncertainty related to the filtering step we included the following lines in the methods (211-214): "However, it should be noted that not for all relatively dry regions only a few samples pass the filter. For example, coastal regions in the northern part of the Mediterranean Basin are relatively dry, yet, a relative large amount of samples passes the filter here. Regions where only a few samples pass the filter are locate in Libya, Lebanon, and Syria. Especially here, results need to be interpreted with care."

Additionally, we included the following lines in the discussion (lines 463-471): "Finally, a relatively large amount of samples is filtered out due to unrealistic model output resulting in uncertainty. This holds specifically for the relatively dry regions where for some grid cells a large fraction of the samples is removed. Nevertheless, in these regions there is a significant amount of grid cells for which 50% or more of the samples pass the filter. It is expected that for a larger number of samples the same percentage of samples will be filtered out, not necessarily

reducing the uncertainty. Due to filtering and uncertainties in ERA5 data, the absolute values shown in Fig. 2 are less meaningful than the spatial patterns. Although the convective rainfall potential (Figs. 3 and 4) is calculated using these absolute values, variations in ABL height, LCL, and CAPE have only a minor effect on its overall spatial distribution (Figs. A8–A10). It should be noted that the aim of this study is not to give an accurate prediction of the hydrological effects of forestation, yet, it aims to identify in what regions forests may contribute to local rainfall."

Furthermore, we included the following lines in the conclusion (lines 559-561): "Soil moisture relates to how the ABL develops over forest and bare soil and also to the uncertainty of the ABL development with higher uncertainty in relatively dry regions."

We believe that these revisions contribute to a clear description of the uncertainty of the findings and improves the studies' transparency. We thank the reviewer for this valuable comment.

Finally, in the methodology we will specify that CLASS is a vertically integrated single column model. We thank the reviewer for pointing this out.

**CAPE Analysis:** Why was a threshold of 400 J/kg chosen for the CAPE analysis? Given that the authors calculate CAPE using the metpy cape\_cin (typo in L186) function, CIN should also be included in the analysis and would provide a more robust standard for determining the likelihood of convective initiation. Recent studies (Emanuel, 2023; Zhang et al., 2023) have also shown that the development of high CIN over wet soils is essential to explaining the development of high CAPE in both models and in observations. Given that the relationship between CAPE and soil moisture is one of the study's main results, the discussion in L364-373 could be expanded and some of that literature should be mentioned earlier in the introduction (L64).

We used the research from Yin et al. (2015) for the design of our study. The study by Yin et al. (2015) reflects on the controls of convective rainfall. One of the controls is a sufficient amount of convective available potential energy. This study describes that for convective rainfall to occur CAPE typically must exceed 400 J/kg. In the revised version of our manuscript we included the following lines to better introduce CAPE (lines 72-76): "Second, whereas the crossing of the ABL and LCL in itself has been considered an indicator of the probability of convective precipitation in previous research (e.g., Juang et al., 2007; Konings et al., 2010), also the convective available potential energy (CAPE) should be accounted for Yin et al. (2015). CAPE is a measure of the amount of energy available for deep convection. For the development of deep convective clouds that can produce rainfall, CAPE needs to be equal or larger than 400 J kg $^{-1}$  ( $\geq$  400 J kg $^{-1}$ ) (Yin et al., 2015). Therefore, to determine the convective rainfall potential we also evaluate CAPE."

Furthermore, we agree with the reviewer that convective inhibition (CIN) would be a valuable variable to include in our study to better understand where deep convective clouds may develop and where this development may be inhibited due to stable layers. Based on the study by Yin et al. (2015) we assume there is a convective rainfall potential when the ABL and LCL cross (ABL $\geq$ LCL), and when there is sufficient CAPE ( $\geq$ 400 J/kg). To include CIN in our analysis, we determine where it exceeds 100 J/kg as deep convection is unlikely for higher CIN values (Wallace and Hobbs, 2006).

We explain the relevance of CIN in the following lines in the introduction (lines 77-81): "Finally, a stable layer or inversion can prevent air to rise and thus reduce convection, which is called convective inhibition (CIN) (Wallace and Hobbs, 2006). CIN represents the amount of energy

that needs to be overcome, e.g., by heating or moistening the air, for convection to occur; a lower CIN allows convective clouds to develop more easily, and deep convection is unlikely for CIN ≥100 J/kg (Wallace and Hobbs, 2006). CAPE cannot be accessed when CIN is too large. In this study, we compare convective rainfall potential with CIN to get a better understanding if deep convective clouds are likely to develop."

We thank the reviewer for this helpful comment. We believe this contribution improves the methodology of our manuscript.

Wallace, John M., and Peter V. Hobbs. Atmospheric science: an introductory survey. Vol. 92. Elsevier, 2006.

Yin, J., J. D. Albertson, J. R. Rigby, and A. Porporato (2015), Land and atmospheric controls on initiation and intensity of moist convection: CAPE dynamics and LCL crossings, Water Resour. Res., 51, 8476–8493, doi:10.1002/2015WR017286.

Spatial Correlation with Soil Moisture Regimes: Given that the interpretation relies heavily on an understanding of the different wet and dry locations in the basin, Figure A1 should be included in the main paper to be more easily accessible. The spatial variations in Figure 2 for CAPE are quite scattered compared to the more coherent patterns in BLH and LCL. Could the authors perform some sort of correlation between the ABL/CAPE changes and the soil moisture variations across the region? That is, how well do the ABL/CAPE changes map onto the soil moisture regimes? Also, given that the study is currently framed in the context of future climate change, please comment on how we expect those regimes to change in the future. If moisture decreases in the region, will these results still hold? Please also discuss how realistic an average  $\Delta$ LCL of over 8km is and rescale the plot in Figure 2.

We thank the reviewer for these valuable suggestions. We included Figure A1 in the main text of our revised manuscript (see Figure 2).

In addition, we studied the correlations between soil moisture and ABL height, LCL, CAPE, CIN and ABL height minus LCL. We did this for both the forest scenario and the bare soil scenario. This resulted in the following table that we included in the Appendix of our manuscript (Table A5).

|                    | Spearman correlation coefficients |        |
|--------------------|-----------------------------------|--------|
|                    | Bare soil                         | Forest |
| BLH                | -0.47                             | -0.50  |
| LCL                | -0.33                             | -0.44  |
| CAPE               | 0.11                              | 0.16   |
| CIN                | -0.08                             | -0.06  |
| Crossing (BLH-LCL) | 0.12                              | 0.17   |

These correlations show that the ABL height (BLH) and LCL correlate to variations in soil moisture across the Mediterranean Basin. CAPE and CIN do not clearly correlate with variations in soil moisture. We explain these results in the following lines of our manuscript (lines 308-317): "The dependency of ABL development on soil moisture is highlighted further by the observed correlations between ABL characteristics and soil moisture content. For both land cover scenarios, the ABL height shows a negative correlation with soil moisture (Tab. A5), indicating the role of soil moisture in modulating the surface energy balance and, consequently, boundary

layer growth. Over the Mediterranean Basin, the LCL also shows a negative correlation with soil moisture, with this relationship being more pronounced over forest than for bare soil conditions (Tab. A5). This suggests that enhanced evapotranspiration in forested areas allows soil moisture to more effectively reduce the LCL, thereby potentially increasing the likelihood of ABL–LCL crossing. Despite this, no consistent correlation is observed between the difference in ABL height and LCL (ABL height minus LCL) and soil moisture (Tab. A5). Similarly, CAPE does not show a clear relationship with soil moisture across the Mediterranean region (Tab. A5), suggesting that a change in the energy balance has a stronger impact on ABL growth than de development of CAPE."

The reviewer points out that a difference in LCL of 8km may not be realistic. A difference in LCL of 8 km is indeed not very realistic. More realistic values would be between 500 and 2500 m. Espy's equation (https://doi.org/10.1016/S0016-0032(36)91215-2) can be used to determine the LCL based on temperature and dewpoint temperature:

$$h_{LCL} \approx 125(T-T_d)$$
.

A height difference of 8 km would mean that there is a difference of  $64^{\circ}$ C between temperature and dewpoint temperature, which is highly unlikely. More realistic values of the difference in LCL would be between 500 m and 1500 m. The unrealistic values of  $\Delta$ LCL are mainly found in the relatively dry regions, yet also in some of the relatively wet regions. The reason for these extreme values is that the calculations in CLASS represent the conditions relatively close to the surface and the LCL is calculated with a constant air density (rho), which results in extreme values when the air is too dry. It should be noted that for the simulations that result in an extreme LCL the potential for significant convection is unlikely due to the dry conditions. Hence, the results in Figures 3 and 4 are likely not affected much by this as for extreme LCL values there is no crossing and thus no convective rainfall potential.

It should be noted however that Figure 2 shows the mean of all simulations for each grid cell. We rescaled the colorbar of this plot to highlight the variation in  $\Delta$ LCL more clearly for the relatively wet regions. In addition, we include the following sentence in the methodology: "The advection fluxes account indirectly for large-scale horizontal atmospheric forcing. The LCL is calculated using constant air density ( $\rho$ ), which results in extreme LCL values under dry conditions. However, it should be noted that under these dry conditions the potential for significant convection is unlikely."

Finally, we discuss in more detail how the future climate may affect the rainfall potential. We included the following lines (lines 501-507): "For example, climate change may reduce local moisture recycling as under drier than normal conditions, local moisture recycling tends to be below average and under wetter than normal conditions local moisture recycling tends to be above average (Theeuwen et al., 2024). In addition, drying due to climate change may negatively affect soil moisture. Due to the negative correlation between soil moisture and ABL height and LCL, climate change may result in deeper boundary layers and higher LCLs over forest and bare soil. However, as LCL shows a stronger correlation with soil moisture over forest than bare soil, drying may have a stronger impact on the LCL over forest than bare soil, negatively impacting rainfall potential.".

We thank the reviewer for their helpful feedback. We believe that these points help to improve our manuscript. Reflecting in more detail on the relation with climate change helps to put our results better in perspective and by including the correlations we are able to underline the main processes better.

**P-E and Moisture Recycling:** Given that the study's goal is to understand "future drying" and the changes in precipitation potential are accompanied by changes in evapotranspiration, there should be some considerations of P-E in the discussion of "wet gets wetter" (Abstract).

This would be an interesting addition and we thank the reviewer for this comment. However, as this method does not allow to quantify the change in precipitation, it is not possible to study how the different land cover types affect P-E across the region. We agree with the reviewer that this information would be highly valuable to better understand whether a forest would contribute to drying or wetting. Because we cannot quantify this we rephrased the aim of our study. Our goal is to assess where in the Mediterranean Basin a mature forest may contribute to more rain locally. We believe that the outcomes of our study contribute to opportunities for future studies as it may give an indication what regions future studies should focus on to determine the hydrological effects of regreening. We clarify our aim in the introduction of our manuscript (line 95). In addition removed the statement "dry gets drier and wet gets wetter" from the abstract. We also removed similar statements from the discussion and conclusion as we agree with the reviewer that based on these results we should not make such statements.

Finally, we would like to thank the reviewer once more for their valuable feedback. We believe these changes helped us to improve our manuscript significantly.

**Detailed response to reviewer comment from reviewer #2**

The reviewer's comments are presented in blue and the authors' response is presented in black. The line numbers refer to the line numbers in the document that includes all the tracked changes.

The study examines the influence of forest cover on convective precipitation potential in the Mediterranean Basin, using the CLASS model to simulate the atmospheric boundary layer (ABL) response to changes in land cover (bare soil vs. forest) and soil moisture. While the paper is well-organized and offers valuable insights into land-atmosphere interactions, there are some major reservations I have regarding the methodology, underlying assumptions, interpretation of results, and the clarity of key definitions that should be addressed and revised in the manuscript before publication.

We are happy to hear that the reviewer believes our manuscript offers valuable insights into land-atmosphere interactions and that they find our manuscript well structured. We are thankful the reviewer took their time to provide us with valuable feedback that helped us to improve the manuscript significantly.

**1. Definition of key terms**

 The terms "forestation," "regreening," and "land restoration" are used throughout the manuscript but lack precise definitions. Please clarify what these terms mean in the context of this paper.

We thank the reviewer for pointing this out, we carefully reconsidered the use of those terms. Instead of using these different terms throughout the manuscript we consistently use the term forestation in our revised manuscript. We introduce this term in the introduction in lines 35-36. Furthermore, taking into account the comments of reviewer #1 we shifted the focus from our manuscript from forestation to forests. We introduce forestation in the introduction to provide the context of our research and in the discussion we connect our results to forestation to

describe the impact of our work. Other than in these two parts of our manuscript we no longer refer to forestation; we refer to forest. We believe this more consistent use of forestation improves the clarity of our manuscript, and therefore, we are thankful for this comment.

• L285: The term "parcels" is introduced for the first time in the Discussion section without prior definition, though it is an important concept in the calculation of CAPE. To ensure clarity, the authors should define this term earlier in the manuscript, preferably in the Methods section, and briefly explain its relevance to the study's atmospheric processes.

The reviewer raises a very good point. We now introduce the term parcel in the methods section in line 241. We agree that this helps to clarify the explanation of the processes in the discussion.

• L55 and L186: Given the study's focus on CAPE and since CAPE is a calculated quantity rather than a height like the ABL or LCL, the authors should explicitly provide the equation used for its computation in the main text. Simply referencing the MetPy Python function is insufficient, as it does not clarify the exact formulation or assumptions applied in this study. Including the full equation and associated assumptions will improve transparency and reproducibility.

We thank the reviewer for this suggestion, we agree that we can more clearly introduce CAPE in our manuscript. In the revision we included the full equation that is used to calculate CAPE. We did the same for CIN, which we also included in the revision. Furthermore, we included a description of these equations and a simple description of CAPE and CIN. We believe that these revisions will clarify the main concepts of our study to a wider audience.

We included the following lines to address this comment (lines 237-253): "CAPE quantifies the potential for deep cloud development and the amount of water that can be condensed, while CIN represents the resistance to cloud formation by measuring how much energy is needed to initiate convection. The crossing of the ABL and LCL describes the potential onset of cloud development.

Near the surface an air parcel (i.e., a small package of air containing water vapor with uniform properties) may be cooler, and therefore heavier, than its environment, naturally resulting in a sinking motion. CIN is a measure for the amount of energy a parcel needs to reach the level at which it can rise freely. If a parcel is adiabatically (without heat transfer) lifted it may become warmer than its environment due to the vertical temperature gradient of the environment. If the parcel becomes warmer, and therefore less dense, than its environment it becomes buoyant and starts to rise. CAPE is the cumulative positive potential energy of a rising parcel that is warmer than its environment. We calculated CAPE and CIN using the cape\_cin function of the MetPy python package (May et al., 2022) that uses the following equations:

$$CAPE = -R_d \int_{LFC}^{EL} (T_{v,parcel} - T_{v,env}) d lnp$$

$$CIN = -R_d \int_{SFC}^{LFC} (T_{v,parcel} - T_{v,env}) d lnp$$

These equations hold under the assumption that the parcel is lifted adiabatically until it reaches the LCL, passed the LCL it rises semi-adiabatically, i.e., condensation leaves the parcel (Wallace and Hobbs, 2006). In these equations  $R_d$  is the gas constant, EL is the pressure at the equilibrium level, LFC is the pressure at the level of free convection, SFC is the pressure at the surface level,

 $T_{v}$  is the virtual temperature either of the parcel or the environment, and p is the atmospheric pressure."

**2. Study setup**

• L57: "When both the ABL and LCL cross and the CAPE is at least 400 Jkg-1 there is convective rainfall potential." The phrasing "both the ABL and LCL cross" needs clarification: does this mean the ABL height exceeds the LCL? Additionally, while CAPE represents atmospheric instability, a threshold of 400 J/kg does not inherently indicate precipitation without considering other key factors such as CIN, mid-tropospheric moisture, and large-scale forcing (Emanuel, 2023). Can the authors clarify the reasoning behind this threshold and account for additional necessary conditions for convective rainfall?

We thank the reviewer for pointing out how we could improve our study set-up and more clearly describe it. First, there is a crossing when the ABL height is equal to or larger than the LCL. We included this statement in several parts of our manuscript using: "ABL  $\geq$  LCL". For example we included this in line 237.

We used the research from Yin et al. (2015) to design our own research. The study by Yin et al. (2015) reflects on the controls of convective rainfall. One of the controls is a sufficient amount of convective available potential energy. This study describes that for convective rainfall to occur CAPE typically must exceed 400 J/kg. In the revised version of our manuscript we included the following lines to better introduce CAPE (lines 72-76): "Second, whereas the crossing of the ABL and LCL in itself has been considered an indicator of the probability of convective precipitation in previous research (e.g., Juang et al., 2007; Konings et al., 2010), also the convective available potential energy (CAPE) should be accounted for Yin et al. (2015). CAPE is a measure of the amount of energy available for deep convection. For the development of deep convective clouds that can produce rainfall, CAPE needs to be equal or larger than 400 J kg $^{-1}$  ( $\geq$  400 J kg $^{-1}$ ) (Yin et al., 2015). Therefore, to determine the convective rainfall potential we also evaluate CAPE."

Furthermore, we agree with the reviewer that convective inhibition (CIN) would be a valuable variable to include in our study to better understand where deep convective clouds may develop and where this development may be inhibited. To include CIN in our analysis, we determine where it exceeds 100 J/kg as deep convection is unlikely for higher CIN values (Wallace and Hobbs, 2006).

We explain the relevance of CIN in the following lines in the introduction (lines 76-80): "Finally, a stable layer or inversion can prevent air to rise and thus reduce convection, which is called convective inhibition (CIN) (Wallace and Hobbs, 2006). CIN represents the amount of energy that needs to be overcome, e.g., by heating or moistening the air, for convection to occur; a lower CIN allows convective clouds to develop more easily, and deep convection is unlikely for CIN ≥100 J/kg (Wallace and Hobbs, 2006). CAPE cannot be accessed when CIN is too large. In this study, we compare convective rainfall potential with CIN to get a better understanding if deep convective clouds are likely to develop."

Finally, we agree that there are more factors that affect convective precipitation. We made several changes in our discussion to highlight this in our manuscript. Line 439-440: "Consequently, we approximated the potential for convective precipitation using CAPE and the crossing of the ABL and LCL overlooking the contribution of mid-tropospheric moisture to convective precipitation." Lines 448-461: "Hence, moisture convergence, which contributes to

the development of convective precipitation, may be underestimated in the CLASS model. Nevertheless, advection of moisture and heat is prescribed in this model, and therefore, horizontal large scale forcing, which also affects convective rainfall, is included in this model to some extent".

We thank the reviewer for this helpful comment. We believe this contribution improves the methodology of our manuscript.

Wallace, John M., and Peter V. Hobbs. Atmospheric science: an introductory survey. Vol. 92. Elsevier, 2006.

Yin, J., J. D. Albertson, J. R. Rigby, and A. Porporato (2015), Land and atmospheric controls on initiation and intensity of moist convection: CAPE dynamics and LCL crossings, Water Resour. Res., 51, 8476–8493, doi:10.1002/2015WR017286.

**L47:** "To isolate the local effects of changes in land cover on local precipitation, a different model approach is necessary." Please clarify what is meant by "a different model approach." Specifically, elaborate on the key differences between this approach and existing methods, and justify why the proposed model is better suited for isolating local effects on precipitation.

What we refer to with "a different model approach" is using a model that solely simulates the local processes that are affected by land use changes. In climate models that use land cover scenarios, a change in land cover upwind from location X may affect precipitation, or other variables, in location X. This makes it challenging to isolate how a land cover change in location X itself may affect the local processes.

To account for this comment, we rephrased the sentence that the author refers to. In lines 61-63 we now write: "To isolate the local effects of changes in land cover on local precipitation a different model approach is necessary; We need to use a set-up that models solely the local processes such that upwind processes do not affect the results"

We thank the reviewer for pointing how we can improve the clarity of our manuscript.

• L83: "This simulation is done for early summer as this is the start of the dry season." It is not clear to me why the early summer (May and June) time period is significant for study when it is stated in L93 that there is late spring and summer convective precipitation in the region. Have the authors considered expanding the period of study (which may also improve the study sampling rate)? Please elaborate on this.

We thank the reviewer for raising this point. During spring/early summer the coupling between the land surface and the atmosphere is stronger compared to other seasons (Ardilouze et al., 2022; Benetó & Khodayar, 2023; Gates & Liess 2001; Lombardo & Bitting 2024). This implies that vegetation relates most to convective precipitation during this time of year. We included a better motivation for the study period in our manuscript in lines 111-113: "This simulation is done for early summer as during this period the coupling between the land surface and atmosphere is stronger than in other seasons (Ardilouze et al., 2022; Lombardo and Bitting, 2024)".

Ardilouze, C.; Materia, S.; Batté, L.; Benassi, M.; Prodhomme, C. Precipitation Response to Extreme Soil Moisture Conditions over the Mediterranean. *Clim Dyn* **2022**, *58* (7), 1927–1942. <a href="https://doi.org/10.1007/s00382-020-05519-5">https://doi.org/10.1007/s00382-020-05519-5</a>.

Benetó, P.; Khodayar, S. On the Need for Improved Knowledge on the Regional-to-Local Precipitation Variability in Eastern Spain under Climate Change. *Atmospheric Research* **2023**, 290, 106795. <a href="https://doi.org/10.1016/j.atmosres.2023.106795">https://doi.org/10.1016/j.atmosres.2023.106795</a>.

Lombardo, K.; Bitting, M. A Climatology of Convective Precipitation over Europe. *Monthly Weather Review* **2024**, *152* (7), 1555–1585. <a href="https://doi.org/10.1175/MWR-D-23-0156.1">https://doi.org/10.1175/MWR-D-23-0156.1</a>.

• L93: "Although precipitation falls predominantly in winter, during late spring and summer there is convective precipitation in the region." Given that the study relies on historical data from ERA5, it would be beneficial to include a figure illustrating which grid cells recorded observed precipitation and which are currently bare soil vs. forest. This would help assess the spatial distribution of convective precipitation and clarify how well the modeled convective rainfall potential corresponds to observed precipitation events given the current state of the region. This would be relevant to strengthen the study's conclusions and recommendation on the need for forestation in different areas to "potentially enhance local rainfall through forestation". Can the authors provide such a figure to support this statement?

We thank the reviewer for this interesting suggestion, we believe that such a figure would be a valuable addition to our manuscript and we included it in the appendix (Fig. A1). We reflect on this figure in the final part of our discussion (lines 545-546): "These elevated regions, and the northern part of the Mediterranean Basin, currently also receive most convective precipitation in the Mediterranean Basin (Fig. A1)."

• Section 2.6 Postprocessing. I am concerned about the disproportionate exclusion of dry regions compared to wet regions due to the filtering process. The bias introduced by the increased sampling size in wetter regions undermines the generalizability of the authors' conclusion that both forestation and an increase in soil moisture can contribute to convective rainfall potential. Given that nearly half of the samples—primarily from dry regions—were excluded, can the authors clarify how this bias affects their findings? Additionally, how do the authors justify applying these conclusions across the Mediterranean Basin area of study when dry regions are underrepresented in the results? (particularly for Fig. 3). Can the authors comment on this?

We agree with the reviewer that our conclusion should be formulated more carefully and that we should explain the implications of the filtering process more clearly. The results have a higher uncertainty in the south of the Mediterranean Basin due to the smaller amount of samples that pass the filter. However, within these relatively dry regions there are quite some grid cells that have 10 (50%) or more samples passing the filtering step (Figure A2). Still, there are some regions where our results have a relatively high uncertainty, i.e., in Libya, Lebanon, and Syria. The results in these regions should be interpreted very carefully.

In the revised version of our manuscript we discuss the bias and uncertainty in more detail. In the methods (lines 211-214) we state the following: "However, it should be noted that not for all relatively dry regions only a few samples pass the filter. For example, coastal regions in the northern part of the Mediterranean Basin are relatively dry, yet, a relative large amount of samples passes the filter here. Regions where only a few samples pass the filter are locate in Libya, Lebanon, and Syria. Especially here, results need to be interpreted with care." In addition, we clarify that a larger amount of samples would not necessarily reduce this uncertainty (lines 207-208): "An increase in samples would not necessarily reduce this bias as it is expected that a similar percentage of samples will be filtered out."

Furthermore, we discuss how the filtering step affects the results in our discussion in lines 463-471: "Finally, a relatively large amount of samples is filtered out due to unrealistic model output resulting in uncertainty. This holds specifically for the relatively dry regions where for some grid cells a large fraction of the samples is removed. Nevertheless, in these regions there is a significant amount of grid cells for which 50% or more of the samples pass the filter. It is expected that for a larger number of samples the same percentage of samples will be filtered out, not necessarily reducing the uncertainty. Due to filtering and uncertainties in ERA5 data, the absolute values shown in Fig. 2 are less meaningful than the spatial patterns. Although the convective rainfall potential (Figs. 3 and 4) is calculated using these absolute values, variations in ABL height, LCL, and CAPE have only a minor effect on its overall spatial distribution (Figs. A8–A10). It should be noted that the aim of this study is not to give an accurate prediction of the hydrological effects of forestation, yet, it aims to identify in what regions forests may contribute to local rainfall."

Finally, we state how this bias affects the conclusion in the final section of the manuscript (lines 559-551): "Soil moisture relates to how the ABL develops over forest and bare soil and also to the uncertainty of the ABL development with higher uncertainty in relatively dry regions."

We believe that these revisions will help the reader to better interpret the results. We are thankful for this valuable comment.

• **Section 2.7 Validation.** The authors mention some numbers to classify "short and tall vegetation cover", but it is not clear to me how these values are retrieved or calculated. Please clarify the source and methodology used to define these classifications.

We thank the reviewer for pointing out this is not clear. These data are obtained from the ERA5 dataset. To clarify this for the reader we included the following text (lines 221-224): "We obtain the short and tall vegetation cover from ERA5. In ERA5, short vegetation includes crops and mixed farming, irrigated crops, short grass, tall grass, tundra, semidesert, bogs and marshes, evergreen shrubs, deciduous shrubs, and water and land mixtures; tall vegetation includes evergreen trees, deciduous trees, mixed forest/woodland, and interrupted forest (Hersbach et al., 2023). However, not all these vegetation types are necessarily found in the study region."

Additionally, we included a figure that indicates where there is forest and bare soil (Fig. A1).

We believe that this addition will clarify our methodology and therefore we thank the reviewer.

Section 2.8 Model output interpretation, L195: "To analyze the uncertainty of the convective rainfall potential we also study the convective rainfall potential for a change in BLH, LCL and CAPE of ±10%." The authors state that they analyze the uncertainty of convective rainfall potential by varying BLH, LCL, and CAPE by ±10%. However, it is unclear why these 10% variations are chosen and if there is any statistical significance towards the conclusion that the "inaccuracy of the exact values may be of less importance" in L344.

These 10% variations are chosen to get more insight into the robustness of the spatial patterns we identified. We were interested in the sensitivity of the model output to small variations in BLH, LCL, and CAPE. However, we were not interested in whether this sensitivity is linear or nonlinear. Therefore, we decided to study the implications of 10% variations. The decision to study 10% variations is somewhat arbitrary, yet it suffices to analyze the relative sensitivity of the model output to the three variables. The statement in line 345 is based on the uncertainty of the

ERA5 dataset. The exact values in ERA5 are subject to uncertainty as mentioned in the official documentation of ERA5

(https://confluence.ecmwf.int/display/CKB/ERA5%3A+data+documentation). The spatial resolution in the ERA5 data has a smaller uncertainty.

The uncertainty in the ERA5 data was the main motivation to study the uncertainty in the spatial patterns. We selected variations of 10% to better understand if a change in either BLH, LCL, or CAPE, due to uncertainty in the exact values of ERA5 would have effects on the spatial patterns that were obtained in our study and whether these patterns are robust despite small variations in BLH, LCL and CAPE. To highlight this in the text we included the following sentence (line 263): "This small variation allows to study the relative sensitivity of the convective rainfall potential to variations in ABL height, LCL, and CAPE, and therefore, the robustness of the results."

We hope this explanation clarifies our decisions for the reviewer and the addition helps to convey this message to the reader.

o It also seems to me that this uncertainty is related to the "inaccuracy of the exact values" and assesses the sensitivity of results to minor perturbations in key variables. How do the authors account for the impact of sampling bias and dataset exclusions (particularly in dry regions) on the robustness of their conclusions (particularly for Fig. 3 and 4 and associated discussion)?

We thank the reviewer for pointing this out. We included a paragraph in our discussion to explain the impact of the filtering step on the uncertainty (lines 463-471). Here we also highlight that changes in ABL height, LCL, and CAPE have a minor effect on the spatial distribution of the convective rainfall potential: "Although the convective rainfall potential (Figs. 3 and 4) is calculated using these absolute values, variations in ABL height, LCL, and CAPE have only a minor effect on its overall spatial distribution (Figs. A8–A10). It should be noted that the aim of this study is not to give an accurate prediction of the hydrological effects of forestation, yet, it aims to identify in what regions forests may contribute to local rainfall." As a result, this uncertainty does not affect our conclusion-where forests may contribute to the convective rainfall potential-much.

We believe that this comment helped us to more clearly describe the implications of the uncertainty on our main conclusions. We thank the reviewer for their valuable feedback.

• For all figures referencing the rainfall potential color scale. Please include in the caption how rainfall potential is defined. Specifically, what constitutes a grid cell to "have a convective rainfall potential" or "have no convective rainfall potential" in the sample?

We thank the reviewer for pointing this out. We included this information in all the relevant captions. For example, the caption of Figure 4 now includes: "The spatial variability of the land cover sensitivity of the convective rainfall potential, i.e., there is both a crossing of the ABL and LCL (ABL>LCL) and sufficient CAPE ( $\geq$  400 J kg-1), if one or both of these conditions are not met there is no convective rainfall potential."

**3. General manuscript proofreading**

 Throughout the manuscript, there are multiple instances of missing commas and periods, which affect general readability and clarity. I recommend a thorough grammatical review to improve sentence structure, punctuation, and overall flow. In particular, some sentences lack necessary commas for readability, and certain sections contain run-on sentences that would benefit from clearer punctuation. See L27, L47, L48 for some (not all) examples. Also, see L186: "cape\_sin" should be corrected to the correct function name "cape\_cin", and L394: "mediterranean" should be capitalized. A careful proofreading by the authors would enhance the manuscript's clarity.

We thank the reviewer for pointing out that the readability and clarity of the manuscript could be improved overall. We conducted a final review by multiple authors in which we paid close attention to the grammar. We thank the reviewer for already pointing out some of the lines which we should revise.

Finally, we would like to thank the reviewer for taking the time to provide valuable feedback on our submitted manuscript. We believe that with the help of the reviewer we were able to make some significant improvements.

**Response to editor's comments**

We thank the editor for taking the time to provide us with valuable feedback. We believe that their feedback helps to clarify some parts of the manuscript. Below the editor's comments are presented in blue and our response is presented in black.

1) Figure 1 is unclear about deltas. Do they go up or down if the profile moves right?

We agree that this should be described more clearly. We include the following line in the caption of this figure: "For the vertical profiles holds: moving towards the right  $\theta$  and q increase." We thank the editor for this comment as it helped us to improve the readability of this figure.

2) Table 2: is soil moisture varying in the scenarios that use ERA5? If so, should there be another comparison between variable and constant soil moisture (with constant land cover)?

We thank the editor for this interesting suggestion. The soil moisture indeed varies across the Mediterranean Basin in the ERA5 scenario. It would be possible to compare the forest scenario directly with the different soil moisture scenarios. However, for each grid cell the change in soil moisture (between the forest scenario with ERA5 soil moisture and one of the soil moisture scenario's) varies across space. We believe this would add complexity to the results which we believe is not necessary for a conceptual study. Therefore, we decided to focus on the differences between the different soil moisture scenarios to determine the role of soil moisture.

3) L154: Should more be said about poor ERA5 data quality? Is there a reason that the extreme value filtering is used at all?

We thank the editor for this relevant comment. If we would not include the filtering step the extreme values would cause an offset in the mean values (Figure 2) and statistics. We do agree that we should elaborate on the uncertainty due to the ERA5 input data. We included the following paragraph in the discussion (lines 463-471): "Finally, a relatively large amount of samples is filtered out due to unrealistic model output resulting in uncertainty. This holds specifically for the relatively dry regions where for some grid cells a large fraction of the samples is removed. Nevertheless, in these regions there is a significant amount of grid cells for which 50% or more of the samples pass the filter. It is expected that for a larger number of samples the same percentage of samples will be filtered out, not necessarily reducing the uncertainty. Due to

filtering and uncertainties in ERA5 data, the absolute values shown in Fig. 2 are less meaningful than the spatial patterns. Although the convective rainfall potential (Figs. 3 and 4) is calculated using these absolute values, variations in ABL height, LCL, and CAPE have only a minor effect on its overall spatial distribution (Figs. A8–A10). It should be noted that the aim of this study is not to give an accurate prediction of the hydrological effects of forestation, yet, it aims to identify in what regions forests may contribute to local rainfall." We hope that these adjustments clarify the uncertainty that is introduced by the ERA5 data. We thank the editor and both reviewers for their valuable feedback on this point. This helped us to improve the manuscript.

**4) Figure 2: are values the mean of the boundary layer?**

We thank the editor for pointing out this was not clearly described in our manuscript. Some variables show indeed the mean value for the boundary layer. We included the following sentence in the caption to clarify this: "LE and H are surface fluxes, theta and q are by definition the mean within the ABL and the RH holds for the top of the ABL."

**5) Figure 3: what is "crossing"?**

We thank the editor for pointing this out. We agree that this should be explained more clearly in the caption. We included an explanation of "crossing" in all relevant captions. For example, in the caption of Figure 4 we wrote: "The spatial variability of the land cover sensitivity of the convective rainfall potential, i.e., there is both a crossing of the ABL and LCL (ABL>LCL) and sufficient CAPE ( $\geq$  400 J kg-1), if one or both of these conditions are not met there is no convective rainfall potential."

---

## Author Response (AR2)

Subject: Rebuttal for manuscript egusphere-2025-289

Dear Andrew Feldman,

Thank you for giving us the opportunity to revise our manuscript "Forest favours conditions for convective precipitation in the Mediterranean Basin". We are grateful for all the time and work the reviewer put into their constructive and valuable feedback and we were happy to hear only minor revisions were necessary. The comments helped us to improve our manuscript.

Based on the reviewers 'comments we made several textual changes. These changes help to improve the transparency and clarity of the manuscript. Second, we included a new plot in the appendix of our manuscript that highlights the dependency of the impact of land cover type on soil moisture content.

We hope that the revisions and clarifications we have made in response to the reviewers' comments have sufficiently addressed their concerns and improved the overall quality of the manuscript. We believe that the revised version is now suitable for publication in *Biogeosciences* and we look forward to your evaluation.

On behalf of all authors, Kind regards, Jolanda Theeuwen

**Reviewers' and editor's comments:**

First, we would like to state that we are very grateful that the reviewers and editor took their time to review our manuscript. We thank the reviewers and editor for their constructive feedback, which helped to improve the quality of our manuscript. Below the comments of the reviewers and editor are presented in blue. The authors' response is presented in black.

However, I have some concerns that the discussion of sample size in Section 2.4 is not appropriately balanced and needs to more accurately represent the results that are being presented. While the added breakdown of the sampling methodology is extremely helpful, it is misleading to claim a sample size of "57,360" when the majority of the figures in the paper are plotting average values for individual grid cells. In Figure 3, for example, the values being shown on the map are CAPE averaged over only 20 days in a 10 year period. While it may be true that the experiments have obtained a total of 57,360 data points, that is not the same as having 57,360 samples for the results being presented. Please be more careful about this.

We agree with the reviewer that we could be more transparent about this. In section 2.4 we included the following lines to improve this transparency (Lines 174-176): "..., resulting- in a total of 20 model runs for each grid cell. The results are calculated for all samples after filtering (see Section 2.6). For the analyses on spatial variation, all samples per grid cell are used."

Line 256: For ten different output variables, the anomalies between the forest scenario and bare soil scenario show the same spatial pattern (Fig. 3) which overlaps with the spatial variability of soil moisture content (Fig. 2)." Throughout the manuscript, the authors refer to "relatively dry regions" or "relative wet regions" between Figure 2 (soil moisture variability) and Figure 3 (anomalies of variables). I wonder if there is a better way to show explicitly differences between these dry or wet regions with their anomaly variables rather than visually comparing between the two figures? I recommend the authors to consider defining different thresholds for dry vs. wet soil moisture from Figure 2 and filtering out these dry vs. wet regions in their anomaly maps in Figure 3 to strengthen their arguments and conclusions within the Results section, especially when they step through each output variable's takeaways.

We agree with the reviewer that it would be valuable to explicitly show the differences between dry and wet regions. To account for this, we included a new figure in the appendix that shows the dependency of the data shown in Fig. 3 on soil moisture. For each of the plots in Fig. 3 we created an xy-plot with on the x-axis soil moisture and on the y-axis each of the variables shown in Fig. 3. This figure, which we include in the appendix as Fig. A8, shows a relationship between soil moisture and the difference between the forest and bare soil scenario for different output variables. In this figure, the title of each plot indicates the spearman rank correlation coefficient and its corresponding p-value. These statistics show for each of the variables a statistically significant relation. These correlation coefficients vary from 0.16 (weak correlation) to 0.61 (moderate correlation).

We introduce this figure in the main text in lines 263-264: "The difference between the forest and bare soil scenario for the different output variables of CLASS all statistically correlate to soil moisture, with the magnitude of the spearman rank correlation coefficients varying between 0.16 and 0.61 (both positive and negative correlations) (Fig. A8)."

Figure A8: The difference between the forest scenario and bare soil scenario for different output variables of CLASS plotted as function of soil moisture content of the top soil layer. Each dot represents one model run. The title of each plot shows the spearman rank correlation coefficient (ρ) and the corresponding p-value. The output that is shown is the latent heat flux (LE), sensible heat flux (H), specific humidity (q), relative humidity (RH), jump in specific humidity at the top of the boundary layer (dq), potential temperature (theta), jump in potential temperature at the top of the boundary layer (dtheta), convective available potential energy (CAPE), boundary layer height (BLH), and lifting condensation level (LCL).

Figure 4 and 5 refer to percentage of samples as a diagnostic to their conclusions between Bare Soil and Forest to conclude that "convective rainfall potential are larger over forest than over bare soil (Line 517)". Can the authors provide more discussion on the significance of these results?

We thank the reviewer for this comment. For Figs. 4 and 5 we see that most grid cells do not have a rainfall potential. The percentage of grid cells that has a rainfall potential is in the order of 1% and 10%. Despite that the number of samples with a rainfall potential may be low, there seems to be a clear relationship between the number of samples with a rainfall potential and the land cover type and soil moisture. This is also supported by the bottom plots in Figs. 4 and 5 that show the spatial variation. These plots show clearly a larger number of grid cells with a rainfall potential over forest compared to bare soil and over the high soil moisture scenario compared to the low soil moisture scenario.

To address this comment, we added the following lines in the results section (lines 312 - 317): "Nevertheless, we see that most grid cells do not have a rainfall potential. The percentage of grid cells that has a rainfall potential is in the order of 1% and 10%. Despite that the number of samples with a rainfall potential may be low, there seems to be a clear relationship between the number of samples with a rainfall potential and the land cover type, i.e., a higher number of samples with rainfall potential over forest than over bare soil. This is also supported by the plot that shows the spatial variation, which indicates a larger number of grid cells with a rainfall potential over forest compared to bare soil."

Line 313: "Additionally, most grid cells have a CIN well below 100 J kg-1 for both land cover scenarios (Fig. A11), suggesting that inversions do not play a major role in preventing deep convection." This sentence is potentially misleading as it attributes the lack of inhibition to the absence of inversions, rather than to their strength. Inversions can exist with varying magnitudes: weaker inversions correspond to smaller CIN values and and convective potential, whereas strong inversions can yield large CIN and effectively suppress convection. Also, for grid cells with rainfall potential, the presence of low CIN is expected, as parcels that overcome the inhibition can access large CAPE. I recommend the authors to consider rewording this statement, and others on CIN in the Discussion section, to clarify that it is the weakness of the inversions (reflected in the small CIN values), rather than their presence/absence, that allows convection to occur.

We thank the reviewer for pointing this out. We agree with the reviewer that this should be rephrased. We rephrased the sentence as follows: "Additionally, most grid cells have a CIN well below 100 Jkg-1 for both land cover scenarios (Fig. A11), suggesting that inversions may be too weak to prevent deep convection."

Line 367: As for most grid cells CIN is small, it is expected that it affects the convective rainfall potential only little." Please specify what "small" CIN means here.

We agree with the reviewer that referring to it a s "small" is unclear. We included "(<100 J/kg)" to clarify what is meant by small.

it would be best to qualify the statement about 400J/kg of CAPE. It is stated with too much certainty and it is unlikely this is always the case given cloud microphysics and other conditions.

We agree that there may be other processes that affect convection such as microphysics. To account for this comment we emphasize that this is an assumption in our study. We included the following lines in the introduction of our manuscript (lines 66-70):

"For the development of deep convective clouds that can produce rainfall, it has been suggested that CAPE needs to be equal to or larger than 400 J kg-1(Yin et al., 2015). However, different processes may affect the amount of CAPE that is necessary to trigger deep convection.

Nevertheless, to determine the convective rainfall potential we also evaluate CAPE and assume that 400 J kg-1 is a sufficient amount of CAPE to trigger deep convection."

Finally, one remaining issue that is still not clear to me with Figures 4 and 5 bottom maps: are these tested conditions the case where all else is held equal? For example, in figure 4, does each grid cell have soil moisture and radiation dynamics similar to its true environmental conditions, but only with the land cover altered? Similarly for figure 5, does the land cover map reflect reality, but only soil moisture is altered? It is a bit challenging to find this in the methods and, in general, these critical details should be stated in the caption.

We thank the editor for pointing this out. Fig. 4 shows the rainfall potential for the bare soil and forest scenario. These results are obtained from the model runs in which only the parameters that are related to land cover are varied. All other variables remain similar to the realistic situation and are obtained (directly or indirectly) from ERA5. Fig. 5 has similar conditions. However, for Fig. 5 the soil moisture content is varied across the different scenarios. As a result for each of the scenarios soil moisture is constant within the study region. Between the different scenarios soil moisture content varies with scenario. The results in Fig.5 are obtained for a full coverage in forest. Hence all parameters related to land cover represent forest. Finally, all other variables remain similar to the realistic situation and are obtained from ERA5.

To clarify this in the manuscript we add the following lines to the captions of Fig.4 and Fig. 5. Caption Fig. 4: "These results are obtained from the model runs in which only the parameters that are related to land cover are varied. All other variables remain similar to the realistic situation and are obtained (directly or indirectly) from ERA5."

Caption Fig. 5: "These results are obtained from the model runs in which soil moisture content is varied among the different cases. Land cover parameters represent the forest scenario. All other variables remain similar to the realistic situation and are obtained (directly or indirectly) from ERA5."